

# Case-to-Case Variability in the Tropospheric Response to Sudden Stratospheric Warmings Revealed by Ensemble Re-Forecasts

Sheena Loeffel[1], Philip Rupp[2], Selina Kiefer[3], Joaquim G. Pinto[3], Thomas Birner[2,1], and Hella Garny[1,2]

[1]Deutsches Zentrum für Luft- und Raumfahrt (DLR), Institut für Physik der Atmosphäre, Oberpfaffenhofen, Germany
[2]Meteorological Institute Munich, Ludwig-Maximilians-University, Munich, Germany
[3]Institute of Meteorology and Climate Research Troposphere Research (IMKTRO), Karlsruhe Institute of Technology (KIT), Karlsruhe, Germany

**Correspondence:** Sheena Loeffel (sheena.loeffel@dlr.de)

**Abstract.** Stratospheric extreme events during Northern winter and spring have been shown to sometimes enhance the subseasonal predictability of large-scale tropospheric circulation patterns such as the North Atlantic oscillation (NAO) and Greenland/European blocking. We aim to quantify the highly variable downward influence of sudden stratospheric warming (SSW) events on the troposphere in numerical simulations. With this aim, we construct a model climatology using the ICON global
numerical weather prediction (NWP) model consisting of possible realistic stratosphere-troposphere evolutions of the coupled troposphere-stratosphere system during winter months. The resulting simulations demonstrate clear stratosphere-troposphere coupling, consistent with observational findings from previous studies. Ensemble re-forecasts, centred around selected SSW events, reveal significant variability in surface responses and robustly show that on average across SSW events, the lower stratosphere serves as a mediator between the upper/mid-stratosphere and the tropospheric flow. We show that the mean tro-
pospheric response to SSWs based on composites and ensemble re-forecasts is heavily case dependent and ties in with the strength of the lower stratospheric anomaly. Our results establish an increased likelihood of developing Greenland blocking with an anomalous lower stratospheric evolution. Moreover, we present indications that the height of wave reflection surfaces can be decisive in establishing persistent lower-stratospheric anomalies and the associated tropospheric response. Overall, we show that individual SSW events differ significantly in their likelihood to induce a canonical tropospheric response, and this
likelihood can be predicted at the onset of the SSW.

## 1 Introduction

Sudden stratospheric warmings (SSWs) are a prominent feature of the Earth's atmospheric dynamics and are known to have a significant impact on tropospheric weather. These changes in the tropospheric circulation are typically characterised by negative anomalies in the North Atlantic Oscillation (NAO) and Arctic Oscillation (AO), resulting in an equatorward shift of
the jet stream and storm tracks in the Northern Hemisphere (see e.g. Domeisen et al., 2020a, for further details). Stratospheric and tropospheric dynamics mostly evolve on substantially different timescales, the former typically within weeks and the latter within days. For this reason, the correlation between tropospheric and stratospheric variability is particularly important for weather forecasts running longer than 2 weeks. Accurately simulating the fundamental principles governing troposphere-





stratosphere interactions can enhance the accuracy of weather forecasts, as well as extend the corresponding time ranges of
predictability.

However, the relationship between surface anomalies and extreme stratospheric events, though clearly observed on average, can vary significantly from case to case. In fact, only a small fraction of events - ranging from two-thirds Karpechko et al. (2017) to one-fifth Runde et al. (2016) - shows a significant surface signal. While Baldwin and Dunkerton (2001) first clearly highlighted that the multi-event composite zonal-mean signature of SSWs propagates downwards and (in some cases) leads
to surface impacts, they also presented phases where anomalous stratospheric conditions do not correspond with noticeable changes in the troposphere. This is also shown in the more recent works of Domeisen and Butler (2020), Karpechko et al. (2018) and Runde et al. (2016). However, Runde et al. (2016) also showed that persistent lower stratospheric anomalies can trigger a consistent response in the tropospheric flow. Despite the well-documented probabilistic surface impacts of these events, the precise pathways through which the stratospheric signal is transferred to (and possibly amplified within) the troposphere remain
poorly understood. Several theoretical pathways nonetheless explain how SSWs can affect surface weather— including remote effects of stratospheric wave driving, planetary wave absorption and reflection, and direct effects on baroclinicity and eddies. Consequently, a wide range of interaction patterns relating to the downward propagation of stratospheric anomalies into the troposphere have been documented (see e.g. Domeisen and Butler, 2020; Domeisen et al., 2020b).

Focussing on individual events, Rupp et al. (2022) analysed the strong polar vortex winter event in 2019/2020 and highlighted
the importance of the lower stratosphere in facilitating the co-evolution of tropospheric and stratospheric extremes. Moreover, the authors stressed the need of large-ensemble simulations to enable a full characterisation of the coupled extremes in the polar vortex and tropospheric jet seen in early 2020. Using forecast ensembles, Kautz et al. (2020) found that the observed SSW in February 2018 and its surface weather impact, e.g. through local blocking phenomena Charlton-Perez et al. (2018), increased the likelihood of colder surface weather. Notably, they were able to partially attribute the occurrence of the 2018 cold
spell to the SSW event.

The inherent variability within the troposphere complicates attribution of circulation anomalies to the stratosphere, especially for individual events. For example, some stratospheric events may be followed by the 'expected' tropospheric response by chance, without actual downward coupling. Likewise, some stratospheric events may involve downward coupling without showing a tropospheric response, because the stratospherically-induced response is overwhelmed by other tropospheric
anomalies. Therefore, probabilistic analyses using large-ensemble simulations, as used in the studies of Kautz et al. (2020), Rupp et al. (2022), and Bett et al. (2023), are a necessary method to quantify the stratospherically-induced anomalous circulation in the troposphere. For this reason, this study employs the ICON global numerical weather prediction (NWP) model to generate a comprehensive set of potential wintertime evolutions of the coupled troposphere-stratosphere system and ensemble re-forecasts based on selected SSW events, as described in Sec. 2. We present our analyses of the SSW composite mean downward
ward coupling in Sec. 3. We use the ensemble re-forecasts to investigate the variability in downward coupling for separate SSW events (Sec. 4) and its influence on the tropospheric circulation (Sec. 5). With our dedicated simulation setup, we aim to clarify whether individual SSW events consistently alter the probability distribution of subsequent tropospheric large-scale circulation patterns, or whether this impact varies significantly from one event to another. Additionally, we evaluate the extent





to which SSWs modify the regional tropospheric flow (Sec. 6) and discuss the underlying factors responsible for differences in
these responses (Sec. 7) before concluding our findings in Sec. 9.

## 2 Model and data set

### 2.1 Simulation setup

We use the ICOsahedral Nonhydrostatic (ICON; version 2.5.0) model of the German weather service (DWD) which runs on a
triangular grid at a horizontal resolution of roughly 40 km. Vertically the model is discretised into 90 terrain-following hybrid-
height levels, with a model top at about 75 km. The output is provided on a $1°x1°$ regular grid on 52 pressure levels at 6-hourly
temporal resolutions. More details on the dynamical core are given by Zängl et al. (2015).

The model runs are structured into two types: We create an 'Event Generating Ensemble' (EGE) to construct a model
climatology, use for statistical analysis and to generate a range of SSW events. We identify 57 SSW events occurring in the
EGE. In a second step, we perform a number of 'spin-off ensembles': these re-forecast the evolution associated with selected
SSWs generated in the EGE. Our approach based on the construction of the set of SSWs obtained from the EGE's internal
variability and of the ensemble-average evolution following selected SSW events, allows us to extract the actual stratospheric
impact against internal tropospheric variability. The following describes in detail how the EGE and ensemble re-forecasts are
designed.

The EGE is created as a free-running 120-member ensemble simulation continuously covering a time period of 8 months
from October to May. We constructed the EGE as a set of three time-lagged 40-member ensembles initialised with realistic
atmospheric conditions taken from Oct. 1st, 2nd and 3rd 2020 and run through to May 31st 2021. Initial conditions were taken
from operational ICON analysis products provided by DWD as individual sets of initial conditions for 40 slightly perturbed
ensemble members. Since we did not aim to model the specific conditions of the winter period 2020/21 but rather winter
periods under average conditions, we used a climatological sea surface temperature (SST) distribution computed based on re-
analysis data (ERA5, see below), rather than the SST distribution provided in the initial conditions. The SSTs are varied daily
following a climatological profile. The model further uses a climatological ozone distribution, provided as part of the ICON
setup. Since the boundary conditions (e.g. SSTs and Ozone) of the EGE run are given as climatological fields, the model will
lose all skill for long lead times and the specific year of initialisation becomes irrelevant. This occurs after about 1 month lead
time (see Section 3, Fig.1). We can therefore interpret each ensemble member as an alternative realisation of the atmospheric
evolution in a winter with fixed boundary conditions. Furthermore, the removal of all inter-annual variability in the boundary
conditions provides a more robust statistical basis to analyse the case-to-case differences of stratosphere-troposphere coupling
after a SSW, than would be given by a (limited) re-analysis dataset. In particular, all differences between SSW events in our
EGE have to result from differences in internal dynamical evolution and cannot be a result of different forcings outside the
mid-latitudes (e.g. ENSO, QBO).

A set of ensemble re-forecast simulations is then performed for 18 selected SSW events identified in the individual members
of the EGE. These are initialised with perturbed initial conditions of the corresponding EGE member at the SSW onset (first





day of negative zonal mean zonal wind at 10hPa and 60° North) or the day before (to minimise resource usage we only store the full atmospheric state every other day) and run for 60 days. Each re-forecast ensemble consists of 40 members, with initial ensemble perturbations constructed following the random field perturbation approach described by Magnusson et al. (2009).

The random field perturbations use the naturally occurring variability patterns within the climatological dataset of our EGE. To perturb a member of a re-forecast ensemble for a specific event occurring in a certain EGE member at day $d_0$, we compute the difference in atmospheric states between two randomly drawn days $d_1$ and $d_2$. Here, $d_1$ and $d_2$ are drawn from two distinct EGE members other than the member containing the event and have to be within a $\pm 15$ day window around $d_0$, to reduce seasonal signals. The difference in atmospheric states at those days is then scaled by a tuning factor adjusted to provide adequate error

growth over the first few weeks of the model run. The method for determining this factor is outlined in Magnusson et al. (2009). Each perturbation pattern then creates two members of the re-forecast ensemble by adding and subtracting it from the unperturbed initial conditions field, leading to an unchanged ensemble mean.

This approach of generating ensemble re-forecasts not only guarantees that each ensemble member of a re-forecast captures the chosen SSW event identically, but also provides the basis for an improved statistical characterisation of the possible tro-

pospheric evolutions following a stratospheric event. The key advantage of this ensemble re-forecast setup can be distilled to the following: While the constructed EGE setup rules out external factors, we cannot deduce from the EGE alone whether the case-to-case variability in the tropospheric response (and lower stratospheric signal) is driven by the tropospheric variability, or by how strongly an SSW event couples downward. This is where the re-forecast simulations play a pivotal role, as the quantification of the distribution of the response is facilitated through the ensemble setup.

We use the ERA5 re-analysis dataset Hersbach et al. (2020) from the European Centre for Medium range Weather Forecasts (ECMWF) as the observational climatological reference when presenting our ICON-generated set of possible wintertime evolutions of the coupled troposphere-stratosphere system in Section 3. The climatology is calculated as the average spanning the years 1980–2019. The ERA5 dataset was obtained as output of the re-analysis on a $1° \times 1°$ regular horizontal grid following pressure surfaces and has a 6-hourly temporal resolution.

## 2.2 Definition of metrics

We identify SSWs as the reversal of the zonal mean zonal wind at 10 hPa and 60°N ($U_{60}^{10}$) and define the SSW central date as the first day the reversal to an easterly wind occurs, following the criteria introduced in Charlton and Polvani (2007). We use the $U_{60}^{10}$ index to encapsulate the strength of SSW events and require the zonal mean zonal wind between two events to recover and remain westerly for 20 consecutive days or more for these to be considered separate SSWs (e.g. Butler et al., 2017). Final

warming events Butler et al. (2015) are excluded from our analyses.

We use standardized geopotential height (GPH) anomalies averaged over the polar cap (60-90°N) as a measure of the large-scale zonal circulation in the stratosphere and troposphere when investigating possible stratosphere-troposphere coupling, capturing the progression and evolution of (downward propagating) signals, both within the stratosphere as well as from the stratosphere to surface level. The geopotential height anomalies are standardised with respect to the 120-member EGE, unless

stated otherwise. We define the surface response metric as the 1000 hPa GPH anomaly averaged over weeks 3-7 post central





date, and define a lower stratospheric response metric as GPH anomalies at 100 hPa. We refer to a SSW as having a lower stratospheric response if the 100 hPa GPH anomaly exceeds $1.5\sigma$ for at least 10 consecutive days, within the first 6 weeks after the central warming date. These time ranges for the tropospheric surface and lower stratospheric response (week 3-7 and first 6 weeks, respectively) are motivated by the timescales typically observed in the downward propagation of stratospheric mean
flow anomalies, spanning days to approx. two months (see e.g. Ding et al., 2023; Scaife et al., 2022; Sigmond et al., 2013). Sensitivity to the choice of time periods in further discussed in Section 5 (see Fig. 9).

To quantify large-scale atmospheric circulation variability we also employ the $\tilde{AO}$ index, which is calculated as the difference in the zonal mean GPH at 1000 hPa between the meridional means over 30–50°N and 60–90°N. As detailed in Rupp et al. (2022), this index serves as a proxy for the strength, latitudinal position and offers the advantage of being computable directly
from model output without requiring a long-term climatological baseline for EOF analysis.

To diagnose wave–mean flow interactions in the stratospheric zonal-mean circulation, we employ the Transformed Eulerian Mean (TEM) formulation Andrews and McIntyre (1976):

$$\frac{\partial \overline{u}}{\partial t} = f_0 \overline{v}^\star + \frac{1}{\rho_0} \nabla \cdot \mathbf{F} + \overline{\mathbf{X}} \tag{1}$$

The Eliassen-Palm (EP) flux $\mathbf{F}$ is defined in terms of eddy momentum and heat fluxes and provides a way to portray the origin
and impact of waves on the zonal mean flow. In Eq.1, the EP-flux divergence appears as an explicit wave-induced forcing term, quantifying the net momentum transfer from propagating waves to the mean flow. Physically, a convergent EP-flux (positive $\nabla \cdot \mathbf{F}$) corresponds to a westward (decelerating) force on the mean zonal wind due to breaking and dissipation of planetary waves, and is largely responsible for decelerating the stratospheric westerly jet. This formalism elucidates the fact that the stratospheric momentum balance is fundamentally controlled by wave drag: stratospheric conditions, such as the strength of
the polar vortex, are strongly coupled to planetary wave activity in the lower stratosphere, consistent with previous findings (see also, e.g. Birner and Albers, 2017; Dunn-Sigouin and Shaw, 2015).

Blocking detection is done for a latitude band of 60°N to 75°N around the Northern Hemisphere using the two dimensional blocking index by Scherrer et al. (2006), resulting in a binary gridpoint-wise blocking occurrence. The index is calculated for every central latitude ($\Phi_0$) between $45°N$ and $75°N$ using the southern and northern GPH gradient at 500 hPa:

$$GHGS = \frac{Z(\Phi_0) - Z(\Phi_s)}{\Phi_0 - \Phi_s} \tag{2}$$

$$GHGN = \frac{Z(\Phi_n) - Z(\Phi_0)}{\Phi_n - \Phi_0} \tag{3}$$

with $\Phi_n = \Phi_0 + 15°N$ and $\Phi_s = \Phi_0 - 15°N$. A given latitude-longitude grid-point is defined as blocked when the following conditions are satisfied: $GHGS > 0$ and $GHGN < -10$ m/°lat. In this paper we show the blocking frequency anomalies (with
respect to climatology) averaged over weeks 3-7 after the onset of the SSW event.

The statistical significance of the difference in means is determined using the one-sided two-sample Student's t-test on a 95% confidence level. We compare the variances of ERA5 and the EGE using a sample-size adjusted Fisher's F-test. To measure the





strength of the linear association between metrics presented in this paper we use the standard Pearson correlation coefficient, denoted by $r$.

## 3  Composite mean downward impact of SSWs in the Event-Generating Ensemble

As detailed in Section 2, the 120-member ensemble simulation provides a large set of possible evolutions of the coupled troposphere-stratosphere system during winter months (see Fig. 1). The ICON climatology of stratospheric polar vortices follows that of ERA5 well (Fig. 1), and the number of SSW events produced by the EGE is similar to that of the ERA5 climatology Butler et al. (2017): Out of the 120 members, 57 developed SSWs; the majority of which occur in January and February (Fig. 1). The EGE controlled conditions suppress differences between ensemble members in terms of interannual variability (e.g. QBO, MJO, ENSO) despite the independent stratosphere-troposphere evolution. We expect a smaller variability in the EGE dataset. Indeed, the variance is indistinguishable from ERA5 after the first month, suggesting an important role of internal variability for the polar vortex. This also suggests that initial condition memory has subsided from November onwards, and one can refer to the individual ICON ensemble members as independent winter evolutions.

The distinct evolution of the stratospheric warming events compared to the modelled climatological progression (Fig. 2a) allows for the potential downward influence of SSWs to be examined: Fig. 2b highlights the significant anomaly in the composite mean tropospheric zonal mean circulation three weeks after the start of the SSW, consistent with observations and previous work (see e.g. Baldwin and Dunkerton, 2001; Lee et al., 2019; Spaeth and Birner, 2022). This tropospheric signal of decreased AO remains statistically significant until five weeks after the SSW onset. Fig. 3 shows the downward progression of the anomalies after weak vortex events in terms of average polar cap GPH. Selecting only events with a lower stratospheric anomaly highlights that the state of an anomalous lower stratosphere influences the strength of the surface response following an SSW event. The lower stratospheric anomalies were classified as anomalously strong if the geopotential height (GPH) anomalies at 100 hPa exceeded values of $1.5\sigma$ for at least 10 days within the 6-week period after SSW onset (see Section 2.2). As shown in Fig. 3b, the SSW events displaying this lower stratospheric signal show a clear increase in the large-scale tropospheric surface anomaly in the weeks following the SSW. Furthermore, the increased surface anomalies differ significantly from the remaining SSW members not satisfying the lower stratospheric threshold.

With the aim of quantifying a possible resulting shift in the probabilities of anomalous tropospheric zonal mean circulation patterns, the probability density functions (PDF) and the distribution of daily data are compared in Fig. 4 for the first 6 weeks post stratospheric event. Consistent with the findings of Baldwin and Dunkerton (2001), we observe pronounced differences in tropospheric anomalies between the composite means of the SSW group ('SSW', yellow) and the EGE members without stratospheric warming events ('No SSW', light grey). The mean tropospheric GPH anomaly for the SSW cluster lies at $0.36 \pm 0.97$ [standardised], whereas the corresponding 'No SSW' group produced a mean value of $-0.2 \pm 0.97$ [standardised]. Comparing the shapes of the probability density functions between these two groups showcases the increase in likelihood of anomalous mean tropospheric zonal mean circulation regimes: values exceeding $+2$ standard-deviations at 1000 hPa are at least twice as likely to occur for the 'SSW'-group, in line with the estimation of the fraction of attributable risk shown in the



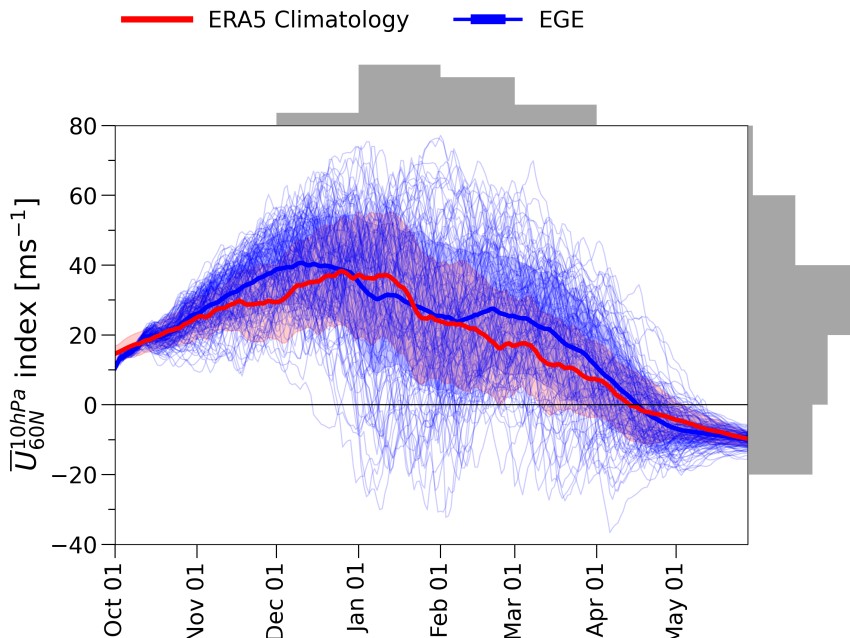

**Figure 1.** Evolution of the zonal-mean zonal wind index $\overline{U}_{60\mathrm{N}}^{10\mathrm{hPa}}$ in ensemble simulations and re-analysis data. Blue: evolution of the EGE for all individual EGE members (thin lines) and ensemble member mean (thick line). Red: ERA5 re-analysis climatology (1980-2019, thick line). The shaded regions indicate the climatological variability (1 standard deviation) of the EGE (blue) and of ERA5 (red). The histograms relate to the EGE simulations (blue) during the winter months only (Dec-Mar). Top panel: Number of SSW events per month, allocated according to the start date of the SSW. Right panel: Distribution of the zonal-mean zonal wind index $\overline{U}_{60\mathrm{N}}^{10\mathrm{hPa}}$.

work of Spaeth and Birner (2022). In particular, values exceeding $-2$ standard-deviations at 1000 hPa are more than twice as likely to occur for the 'No-SSW'-group. Furthermore, the increased tropospheric anomaly of the 'LS-signal' cluster previously shown in Fig. 3b is clearly visible in Fig. 4a. in the form of a distinct right shift of the PDF (upper panel) as well as in the box-and-whisker plots (bottom panel). This shift reaches its peak four weeks after SSW onset (not shown). Surface anomalies
exceeding $+2$ standard-deviations are more than four times more likely than the 'No SSW'-regimes, while the ensemble members experiencing a SSW but no lower stratospheric signal are at least twice more likely (green). In fact, the latter group (SSW without lower stratospheric anomaly) behaves similarly to the modelled climatology with respect to composite means and tail ends of the respective PDFs. In terms of the lower stratospheric evolution following a SSW event, positive anomalies are more likely to occur in the lower stratosphere after a SSW event takes place, as can be seen by the distributional shift shown in Fig.
4b.

Overall, our analyses of the EGE show a downward influence of SSW events on the tropospheric ensemble-mean flow. This is consistent with previous studies on stratosphere-troposphere coupling, e.g. Rupp et al. (2022); Runde et al. (2016), some



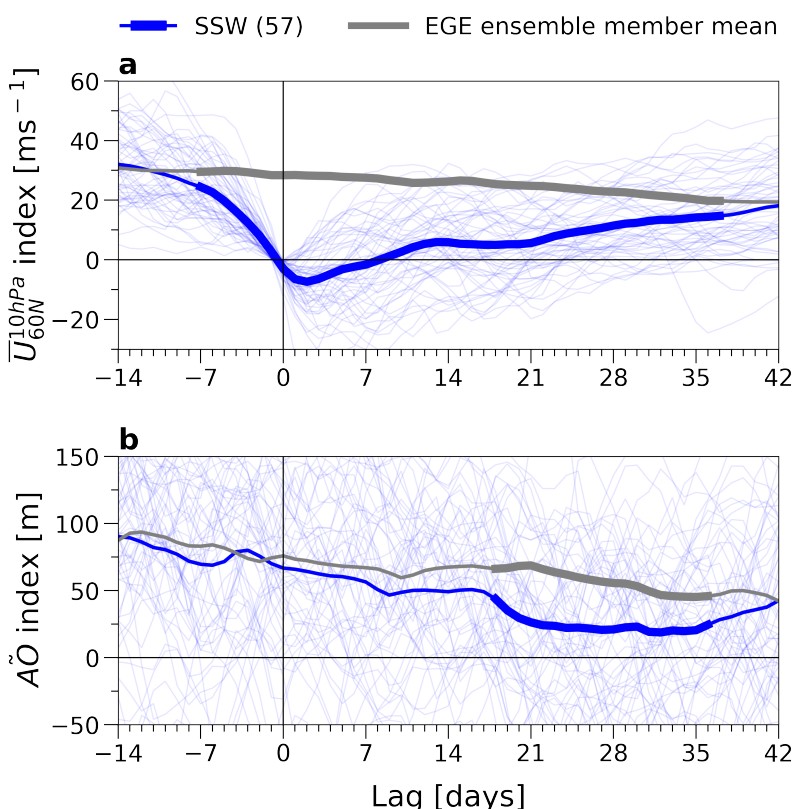

**Figure 2.** Lagged timeseries of the (a.) zonal mean zonal wind at 60°N/10 hPa and (b.) AO index for the EGE climatology (thick, grey) and SSW composite (thick, blue). The number of EGE members in the SSW composite is shown in brackets, and the individual members are shown with the thin blue lines. Note that the climatology is calculated as the ensemble member mean of the EGE. The thickened line segments indicate statistically significant difference in means between the composite SSW and climatology (to 95% confidence). Day 0 is taken as the start of the SSW, ie. the first day the zonal mean zonal wind at 60°N/10 hPa becomes negative, and is marked with the thin black vertical line.



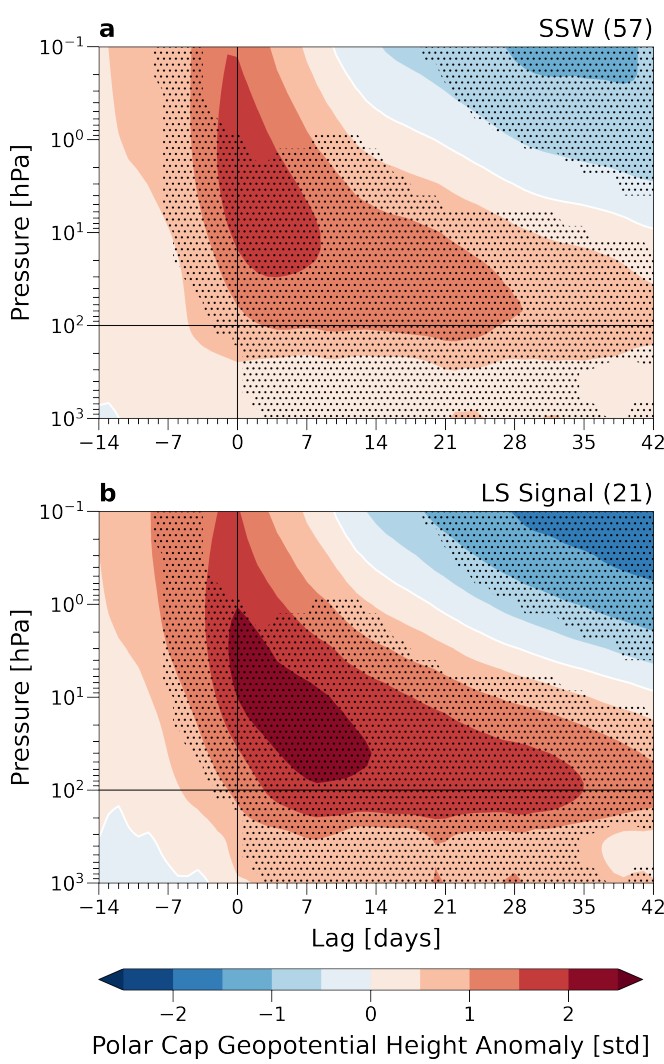

**Figure 3.** Event-based composite of standardised geopotential height anomalies over the polar cap region (60-90°N) of (a.) all SSWs identified in EGE and (b.) only the events with a strong lower stratospheric (LS) signal following the SSW. Numbers in the panel headings represent group sizes. Stippling indicates significant differences (to 95% confidence) in cluster means between the LS signal cluster and members without LS signal (not shown). Thin black lines in the vertical and horizontal mark the start of the SSW and 100 hPa pressure level, respectively.



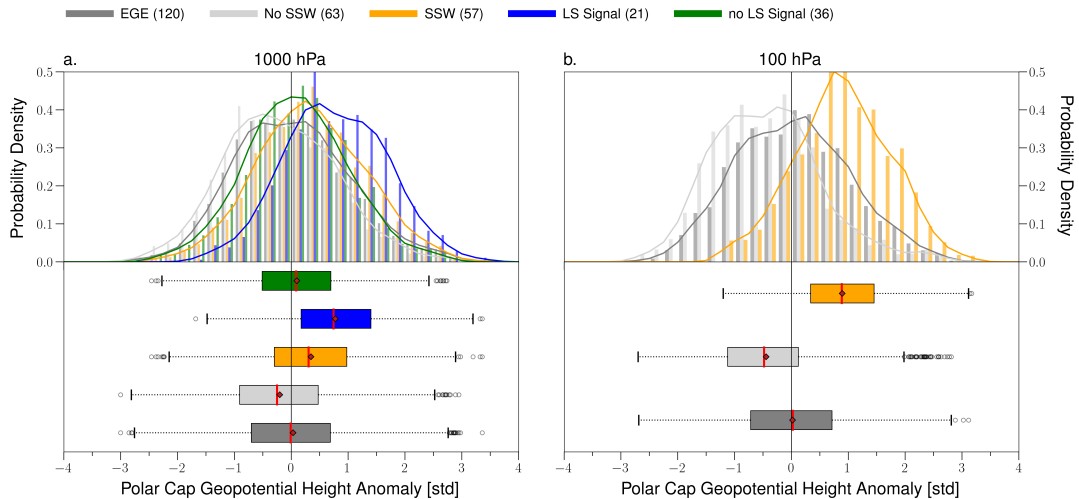

**Figure 4.** Probability density functions (top panel) and distribution (lower panel) of polar cap geopotential height anomalies for weeks 3-7 post event at (a.) 1000 hPa and (b.) 100 hPa from the EGE simulation. Dark grey: climatology, light grey: members without SSWs, yellow: with SSWs. Of the SSWs, members with and without a lower stratospheric (LS) signal are shown in blue and green, respectively. Mean and median shown as red diamonds and vertical lines, respectively. Number of members per group is shown in brackets. Note that for the climatology and No-SSW groups, a random day in January or February was selected as the event date. We present the distribution of surface and lower stratospheric anomalies using boxplots, set the interquartile range as the range between the 25th and 75th percentile and calculate the minimum and maximum values as 1.5 times the interquartile range below the lower and above the upper quartile, respectively. Any data point exceeding these values are classed as outliers.

of which classified stratospheric extreme events according to their downward propagation to the troposphere. Our analyses in particular highlight the important role of the lower stratosphere for downward coupling, pointing towards the lower stratosphere

acting as a mediator between the upper/mid-stratosphere (SSW event) and the troposphere. Based on our findings from the composites of SSWs, we utilize the ensemble re-forecasts in the next Section to examine whether the development of a lower stratospheric anomaly is more or less likely to develop for individual SSWs, and whether this determines the likelihood for a tropospheric response to develop.

## 4 Selecting SSW events for Ensemble Re-forecasting

The selected re-forecast events cover the range of possible stratospheric SSW evolutions (also in terms of extremity) throughout the winter months from December to February. For each re-forecast, the initial stratospheric conditions in terms of the $U_{60}^{10}$ index on the day of the selected event are shown in Fig. 5.

In addition to covering a broad range of initial states in the stratosphere, the SSW re-forecast simulations also capture the range of possible coupled stratosphere-troposphere evolutions (Fig. 6). As expected, the majority lie in the lower right quadrant,





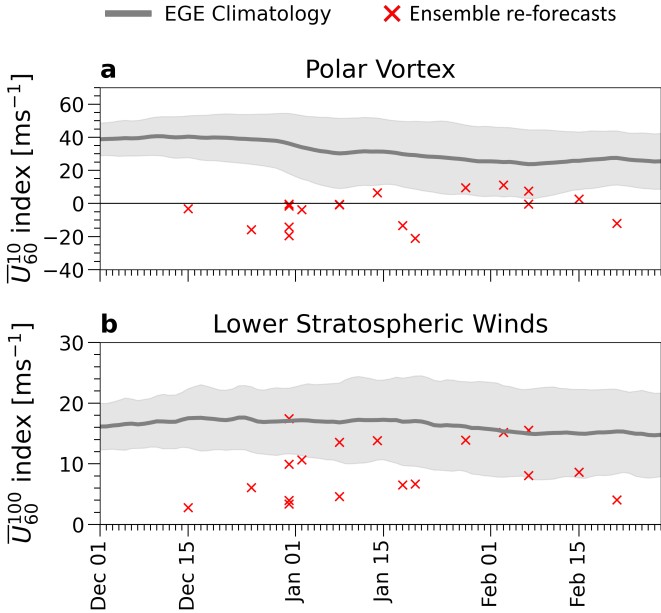

**Figure 5.** Overview of initial stratospheric conditions in terms of a. polar vortex strength and b. lower stratospheric winds of the ensemble re-forecast simulations based on SSW events (red crosses) in the winter months from December to end of February. The thick line represents the EGE ensemble member mean, with 1 standard deviation shown as shading.

as observational data has shown that SSW events are linked to increased surface GPH anomalies (i.e. negative anomalies on the y-axis and positive anomalies on the x-axis; see e.g. Kautz et al. (2020)). The upper left and right quadrants capture weak warming events that are followed by negative or positive (i.e. weak to strong) tropospheric signals, respectively. Figure 6 further contrasts the selected EGE members with the ensemble mean of the 40 ensemble re-forecasts (coloured crosses and dots). Note that for each selected event, the selected EGE member represents one of the 40 members of the re-forecast and the

initial stratospheric state is very similar: The deterministic state lasts up to 10 days and the state is shown here averaged over the first 14 days. We see a greater difference in the values from the selected SSW of the EGE and the re-forecast ensemble mean in terms of the tropospheric surface response index, as this index captures the surface state several weeks after initialisation once all members have evolved separately, which again emphasises the key advantage of the ensemble re-forecast simulation setup.

## 5 Differences in downward coupling in selected SSW events

In Section 3 we showed that our ICON EGE simulations reproduce the known downward influence of SSW events on the tropospheric flow and established that on average across SSW events, a persistent reduction of the zonal circulation in the lowermost stratosphere following the SSWs increases the likelihood for tropospheric flow anomalies to occur. In this section





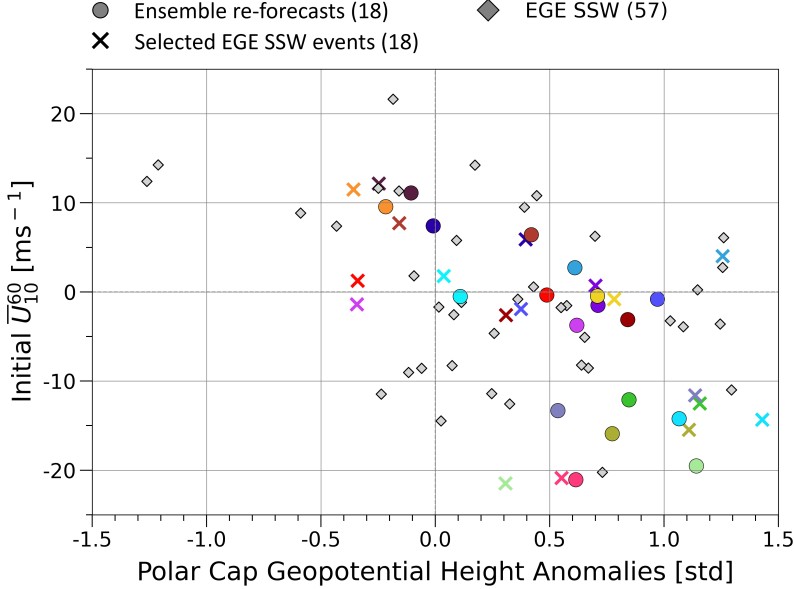

**Figure 6.** Overview of selection of SSW re-forecasts (circles, different colours) in terms of the zonal mean zonal wind at 10 hPa, 60°N averaged over the initial two weeks and the tropospheric surface response index in the weeks after SSW onset. The 57 SSW events from the event-generating EGE are shown as diamonds. Of these, the EGE members selected for the initialisations of the re-forecasts are marked separately with crosses and are shown in the corresponding re-forecast colour. The number of members per group is shown in the brackets.

we analyse if the forced SSW impact onto the troposphere is strongly related with the 100 hPa stratospheric anomaly, and whether it is case dependent.

Fig. 7 presents the distribution of the polar cap geopotential height anomalies 3 to 7 weeks after SSW onset at the surface (Fig. 7a.) and lower stratosphere (Fig. 7b.) for all SSW re-forecasts. The results at 1000 hPa illustrate the varyingly positive surface responses in the ensemble mean for the SSWs, in line with Fig. 6. For this reason, we can conclude that across individual SSW events, some are more likely than others to develop a tropospheric response in the aftermath of the stratospheric event. This is an important result because it suggests that some observed SSW do show the canonical tropospheric response while others do not, which might not entirely be due to internal tropospheric variability (i.e., by chance). Rather, there is predictive power at the day of the SSW on the specific likelihood of this SSW to develop a response.

One might suspect that stronger wind reversal events in the stratosphere have stronger surface impacts, and vice versa, but this is not generally the case as illustrated in Fig. 8, depicting polar cap geopotential height standardised anomalies. While the ensemble re-forecast simulation 'SSW 1' captured both a strong warming event as well as a strong surface response in the ensemble mean (Fig. 8a and d, respectively), the correspondingly weak event in ensemble re-forecast 'SSW 3' (Fig. 8, third column) shows no evidence of downward coupling in the ensemble mean. Another equally weak SSW event in terms of a stratospheric reversal of winds ('SSW 2', middle column) develops positive surface anomalies of comparable magnitude with those of the strong event 'SSW 1'. While one could first assume this difference relates to the strength of the SSW, a clear





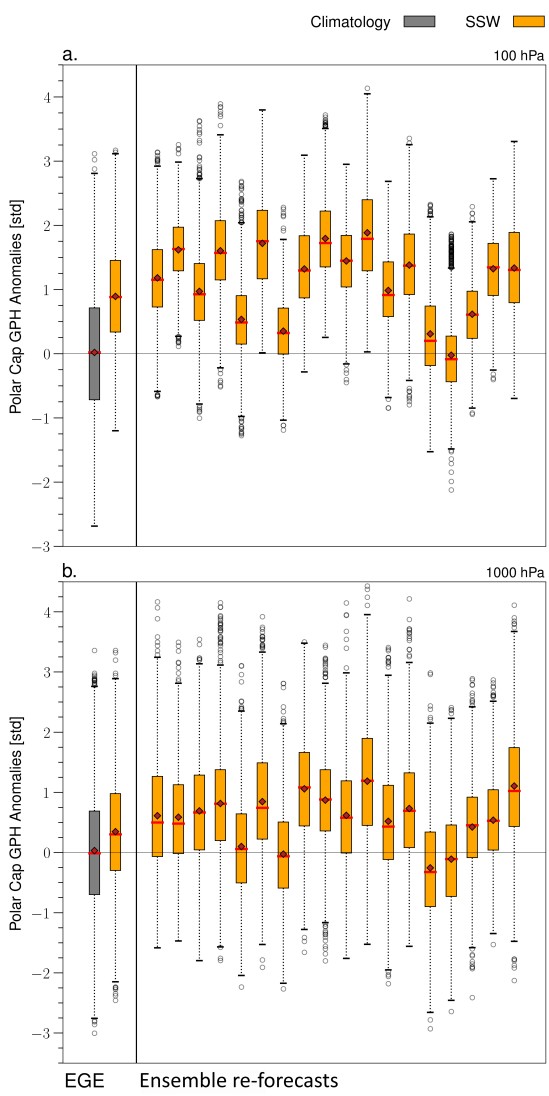

**Figure 7.** Distribution of standardised geopotential height anomalies over the polar cap at (a.) 100 hPa and 1000 hPa (b.), selecting daily values from weeks 3-7 post-event. Within each plot: The left panel shows values for the EGE as shown in in Fig. 4, the right panel shows all re-forecast simulations. Colours are as in Fig. 4: the EGE ensemble mean (ie. EGE climatology) is shown in dark grey. Yellow depicts the composite mean of the EGE SSWs (left panel) and the re-forecast ensemble mean (right panel). Mean and median shown as red diamonds and vertical lines, respectively. We present the distribution of surface and lower stratospheric anomalies using boxplots, set the interquartile range as the range between the 25th and 75th percentile and calculate the minimum and maximum values as 1.5 times the interquartile range below the lower and above the upper quartile, respectively. Any data point exceeding these values are classed as outliers.





link between the event strength and surface response is visible only to a limited extent (see also e.g. Fig. 10b). However, the
evolution of the lower stratosphere points towards the importance of an anomalous state at 100 hPa for a surface signal to
develop (compare 'SSW2' to 'SSW3'). Moving beyond these three examples in Fig. 8 and considering the full set of ensemble
re-forecast simulations, we indeed observe that the ensemble mean response in the lower stratosphere does differ substantially
between the SSW events, cf. Fig. 7b. We corroborate this with Figure 9, in which the ensemble mean GPH anomaly at different
times and heights is correlated against the ensemble mean surface response (averaged over week 3-7) across the 18 re-forecasts.
Here, a weaker correlation of the surface response with GPH anomalies at 10 hPa than with those at 100 hPa is found. Most
notably, the surface response can be considered highly correlated with the geopotential height anomaly at 100 hPa from week
2 onwards, as the composite correlation coefficient exceeds 0.8 and approaches maximum values. This correlation peak in the
lower stratosphere appears before the surface signal develops.

Previously we established that the occurrence of a 100 hPa anomaly plays a key role in whether we see downward coupling
across the SSW-composite mean (see Fig. 4). Fig. 10 now summarizes and underlines the relationship between the state of
the lower stratosphere and the strength of the tropospheric surface signal across events. The lower stratospheric geopotential
height anomalies display a particularly strong relationship with the surface anomalies, with the Pearson correlation coefficient
at 0.85 (Fig. 10). In comparison, the relationship between the surface anomalies and initial vortex strength bears a negative
correlation coefficient of 0.71, as shown in Fig. 10b. Consequently, the surface response is more strongly correlated with the
lower stratospheric state post-event than with the strength of the initial event itself. Our findings shown in Fig. 10 combined
with Fig. 9 allow us to further specify that the 100 hPa anomaly characteristic of a given SSW two weeks after the initial event
strongly correlates with the surface outcome. Given that the strength of the lower stratospheric anomaly is not uniquely linked
to the SSW strength, the question arises which dynamical mechanisms lead to the development of the lower stratospheric
anomaly. This is examined in Section 7.

## 6  Differences in the regional tropospheric impact in selected SSW events

Subsequent to the analysis of the zonal mean, regional effects are examined to assess whether the differences in zonal mean
downward coupling between events are reflected in variations in regional impact. Both the EGE and ensemble re-forecast
simulations produce a surface response pattern consistent with the well-documented general surface response to SSW events
(see e.g., Baldwin and Dunkerton, 2001; Charlton-Perez et al., 2018; Karpechko et al., 2018). This is evident in Fig. 11b-e,
showing close to zonally symmetric, positive mean sea level pressure anomalies (MSLP) over the polar cap in weeks 3-7 after
SSW onset. Notably, the composite of SSW events from the ensemble re-forecast simulations (Fig. 11c) not only exhibits
slightly stronger anomalies over the polar cap but also shows enhanced statistical significance. This key advantage of the
ensemble re-forecast approach presented in this study becomes evident when comparing overall statistically significant regions:
the ensemble re-forecast composite (18 events) covers a broader area and exhibits stronger signals than the composite derived
from an equal number of selected EGE events. The 18 re-forecast SSW events are further classified based on the anomalous
state of the lower stratosphere two weeks after SSW onset, using the 100 hPa GPH anomaly threshold of $1.5\sigma$ (cf. Fig. 9 and





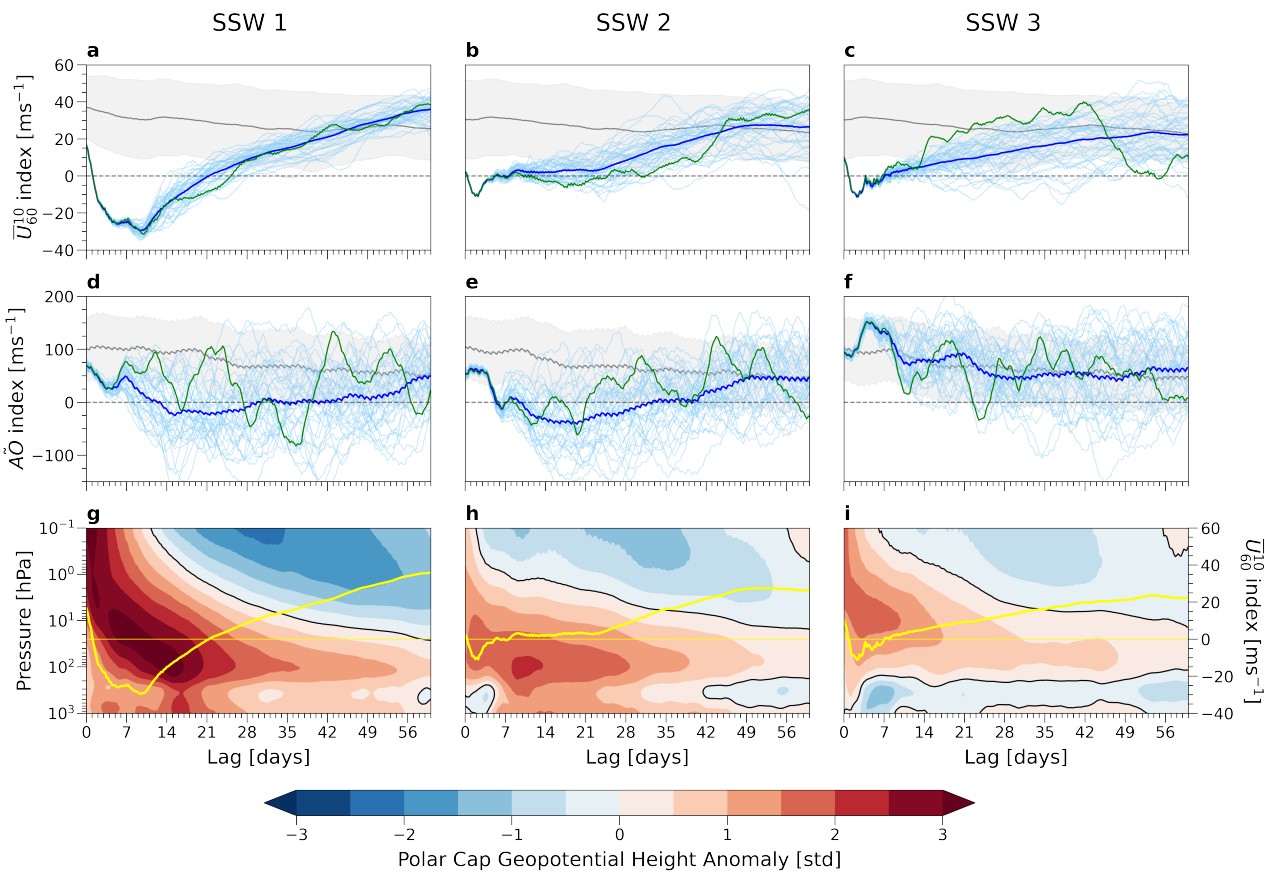

**Figure 8.** Time series of zonal mean zonal wind at 60°N, 10 hPa (a-c) and AO index (d-f) of selected ensemble re-forecast simulations, shown as columns. a-f: re-forecasts (blue) are based on selected SSW events from the EGE (control, green). EGE ensemble mean and standard deviation are shown as the grey line and shading, respectively. g-i: standardised geopotential height anomalies over the polar cap region (60-90°N). The ensemble member mean of the zonal mean zonal wind at 60°N, 10 hPa has been added as yellow lines. Day 0 is taken as the start of the SSW.



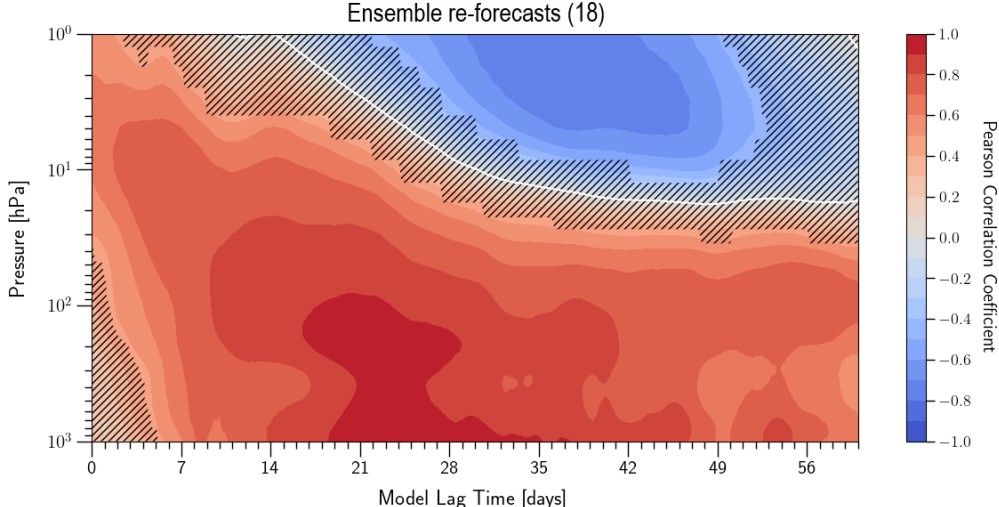

**Figure 9.** 'Dripping paint' plot showing the ensemble re-forecast composite Pearson correlation coefficient, calculated from the respective GPH anomalies correlated with the 1000 hPa GPH anomalies averaged over week 3-7. All GPH anomalies calculated over the polar cap region. Stippling indicates regions of statistical insignificance with p-values exceeding 0.05 (to 95% confidence).

Section 2). The surface signal intensifies for events associated with a stronger lower stratospheric state (Fig. 11d), both in terms of signal strength as well as the size of the region covered.

In both ERA5 (Fig. 12a) and ICON (Fig. 12b-e), atmospheric blocking frequency in weeks 3–7 following SSW onset shows positive anomalies over Greenland and surrounding regions. The re-forecast simulations capture statistically significant increases in Greenland blocking, particularly around the Davis Strait, Greenland, and Baffin Island, with frequency values reaching 0.05 (approximately 4 days per season). The enhanced statistical significance in the re-forecast SSW composite (Fig. 12c) further underscores the benefits of this simulation approach. Analogous to Fig. 11d, the blocking signal strengthens when selecting SSW events by lower stratospheric anomalies: cases with strong anomalies show a more than twofold increase in blocking frequency over Greenland compared to the re-forecast composite, exceeding 0.1 (about 9 days per season) in Fig. 12d.

Given the statistically significant Greenland blocking signal, we compute the correlation between large-scale zonal mean surface anomalies and regional Greenland blocking frequency, finding a strong correlation of 0.84 (Fig. 13b). The correlation between lower stratospheric (100 hPa) anomalies and blocking frequency is slightly lower but remains distinct (0.71). Notably, ensemble re-forecast events with weak lower stratospheric anomalies ($< 1.5\sigma$, blue shading) and strong anomalies ($> 1.5\sigma$, red shading) show an almost full separation (see Section 8 for more details). These results highlight the strong connection between large-scale zonal mean flow and regional blocking variability, though causality remains uncertain. Individual EGE events do not produce any correlation (see Supplement), which emphasizes that an ensemble is needed to bring out the response.

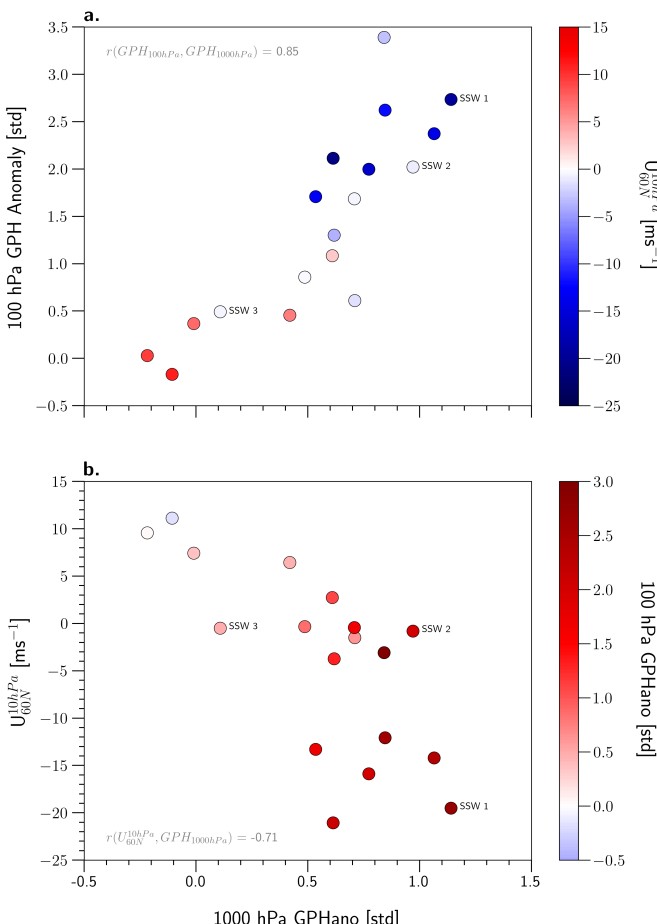

**Figure 10.** Scatter plot showing the correlation of 1000 hPa GPH anomalies over the polar cap region with (a.) 100 hPa GPH anomalies over the polar cap region and with (b.) initial 2-week average polar vortex strength for the ensemble means of all 18 ensemble re-forecasts. Each ensemble mean point is shaded according to the initial 2-week average polar vortex strength (a.) or polar cap GPH anomalies at 100 hPa (b.), as shown with the colour bars. Values of the correlation coefficient: (a.) between 1000 hPa and 100 hPa GPH anomalies: 0.85. (b.) 1000 hPa GPH anomalies and initial vortex strength: -0.71



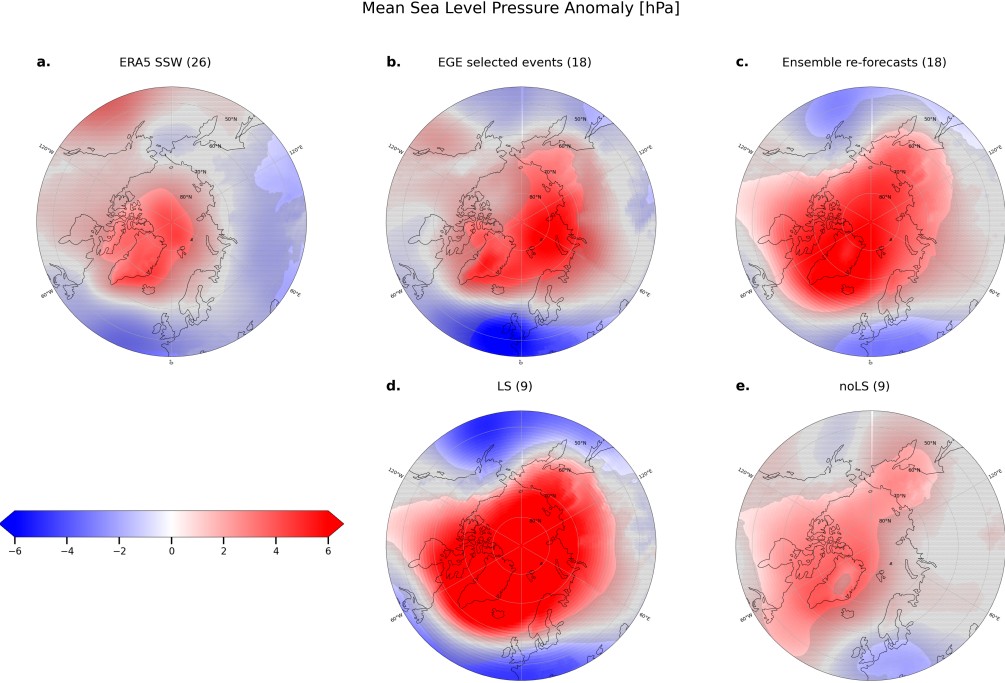

**Figure 11.** Mean sea level pressure anomalies averaged over weeks 3-7 after onset for composites of a. ERA5 SSW events, b. selected EGE events (18), c. ensemble re-forecast events (18), d. re-forecast SSWs with lower stratospheric anomaly (LS), and e. without lower stratospheric anomaly (noLS) development in week 2 after event onset. Stippling indicates regions not of statistical significance (to 95% confidence). Stereographic projection of latitudes north of 45°N, centred on 90°N with 10° graticule.

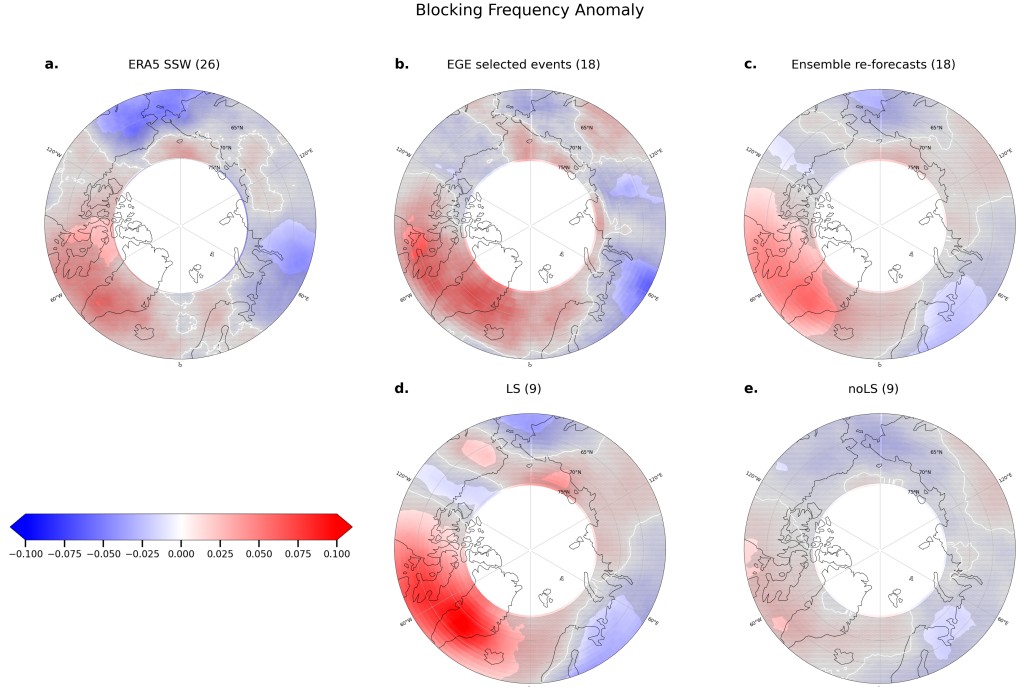

**Figure 12.** Blocking Frequency Anomalies, calculated as the average number of blocking days in the week 3-7 average after SSW onset. Shown are the anomalies averaged as composite SSW events in a. ERA5 re-analysis data, b. selected EGE events, c. ensemble re-forecast simulations, and ensemble re-forecast simulation members with and without strong lower stratospheric anomalies post-event (d. and e., respectively). Numbers in parenthesis indicate number of members per composite. Note that the anomaly is calculated as the difference to the corresponding climatological group (e.g. for a. the anomaly is calculated as the ERA5 climatology subtracted from the average number of blocking days per ERA5 SSW event). Stippling indicates regions not of statistical significance (to 95% confidence).





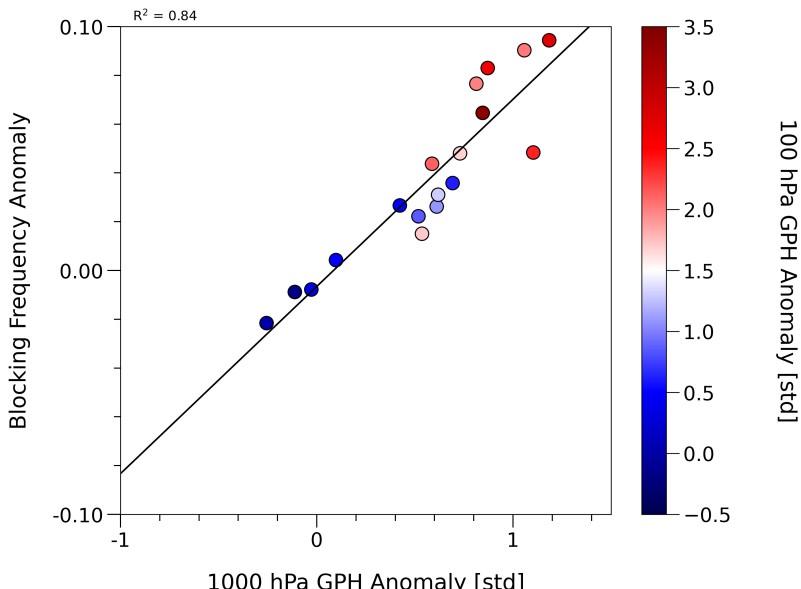

**Figure 13.** Scatter plot showing the correlation between ensemble member mean 1000 hPa GPH anomalies and total number of blocking days for the 18 ensemble re-forecast simulations over the region 60-75°N, -90–30°E. Each point is shaded according to the polar cap GPH anomalies at 100 hPa in week 2 after SSW onset, see colour bar).





## 7 Towards a mechanistic understanding of why not all SSW events lead to lower stratospheric anomalies

As presented in Section 5, the evolution of an anomalous state in the lower stratosphere is instrumental in whether a surface signal develops. We further note that the lower stratospheric zonal mean zonal wind (specifically at 100 hPa) correlates very strongly ($r = -0.91$) with the lower stratospheric geopotential height anomalies at 100 hPa (not shown). Since the stratospheric evolution after a warming event is controlled by the interaction of planetary waves and the zonal mean flow, we investigate the momentum budget of the transformed Eulerian mean (TEM) equations in order to understand the development of a flow

anomaly (or lack thereof) in the lower stratosphere.

In Section 5, we presented three ensemble re-forecast simulations: one with both a strong warming event and surface response in the ensemble member mean (SSW 1, Fig. 8), and two weak SSW events in terms of wind reversal, whereby one showed evidence of downward coupling in the ensemble member mean (SSW 2) and the other did not (SSW 3, Fig. 8). To provide insight in the stratosphere-troposphere evolution following a warming event, key terms for these selected simulations

stemming from the TEM momentum budget equation and Eliassen-Palm (EP) flux anomalies are shown in Fig. 14. These terms include the temporal changes in the background zonal flow (i.e. total wind tendency) and disturbances by large-scale waves, specifically, the contribution of the wavenumber (WN) decomposed EP-flux divergence (WN1 and WN2). We additionally present the anomalies of the EP-flux WN1 and WN2 components. Hatched areas indicate regions where all ensemble members exhibit positive EP-fluxes, corresponding to zones of wave reflection. The meridional residual circulation term from the TEM

momentum budget ($f_0 \overline{v}^\star$, see Eq. 1 in Sec. 2.2)) is not shown, as it effectively balances the combined contributions from the WN1 and WN2 EP-flux divergence.

Focussing on the zonal mean zonal wind tendencies shown in Fig. 14a-c, we observe for the strong event SSW 1 (Fig.14a) the initial deceleration at 10 hPa propagates downwards in typical fashion, and a resulting flow anomaly arises at 100 hPa in the first two weeks. Regarding SSW 2 (Fig.14b), the zonal mean zonal wind also decelerates at 100 hPa, albeit less pronounced.

In comparison, the initial deceleration in the other weak event SSW 3 (Fig.14c) with weak surface response (cf. Fig.8) is interrupted after a few days following the reversal of winds, and in its place, a relatively strong acceleration develops.

Moving on to the interaction of planetary waves and the zonal mean flow (Fig. 14d-o) we notice for SSW 1 the strong initial upward wave activity flux in WN2 (Fig.14m) matches the deceleration due to EP-flux divergence in the upper stratosphere, whereby the latter reaches the lower stratosphere (100 hPa) by the end of the first week at lag day 7 (Fig.14g). We also

observe wave reflection in WN1 constrained to the upper stratosphere only throughout the first two weeks, reaching the lower stratosphere (and below) by day 7 (Fig.14j). While the corresponding acceleration in the upper stratosphere (Fig.14d) does roughly match the pattern of wave reflection, the strengthening of winds does not reach stratospheric levels below the 50 hPa level. Overall, the initially decelerated winds in the lower stratosphere only increase slowly, with a weakly positive zonal mean zonal wind tendency for the remaining run time at 100 hPa (Fig.14a). Thus, lower stratospheric anomalies can develop in this

case due to the uninterrupted deceleration by downward progressing planetary wave forcing, making the occurrence of strong surface anomalies far more likely.

The re-forecast capturing event SSW 2 – with similarly weak reversal of winds but a strong surface response comparable with





the strong SSW 1 event (cf. Fig. 8g) – presents initial (and dominant) strong deceleration exerted by the EP-flux divergence in WN1 (Fig.14e), which goes hand in hand with positive anomalies visible in the upward EP-flux WN1 component (Fig.

14k). There are also regions of WN1 wave reflection, but limited to the initial week in the upper stratosphere (corresponding to the reversal of winds) and ensuing weeks show (moderate) anomalous upward wave activity. However, note that the lower stratospheric wave activity in the first two weeks in WN2 consists of wave reflection, spanning tropospheric and stratospheric (up to 30 hPa) pressure levels (Fig.14n). Correspondingly, the acceleration (resulting from the wave reflective surface) seen in Fig.14h occurs at levels around the reflection surface at 30 hPa, and the deceleration of winds at 100 hPa is predominated

by WN1, allowing for the lower stratospheric response to evolve. In other words, overall the lower stratospheric winds do not develop remarkable acceleration despite persistent reflection of WN2 and consequently, anomalies in the lower stratosphere forced by WN1 can persist.

Meanwhile, the weak event SSW 3 shows initial strongly anomalous upward EP-flux in WN1, exceeding $2\sigma$ (Fig.14l), that corresponds to the initial deceleration in the upper stratosphere (Fig.14c). The deceleration in WN1 (Fig.14f) propagates down

to the lower stratosphere (100 hPa) during the first week (lag day 7), and is followed by acceleration at lower stratospheric levels thereafter. This pursuing acceleration in WN1 can be explained by the areas of negative EP-flux as illustrated by the hatchings in Fig.14l: while initially visible in the troposphere starting mid-week1, planetary WN1 waves are further reflected in the lower stratosphere (particularly between 100-50 hPa) in the 2 weeks that follow. As a consequence, the lower stratosphere can neither enter nor continue in an anomalous state (c.f. Fig. 8i) and the possibility of developing a surface response in the ensuing weeks

becomes much less likely. Note that for WN2, the first two weeks also show acceleration of the lower stratospheric winds resulting from EP-flux divergence in WN2 (Fig.14i), which goes hand in hand with the observed wave reflection in the first week in the troposphere and lower stratosphere (Fig.14o).

Based on the examples examined here, we hypothesise that the height of the wave reflection surface plays a paramount role in whether an anomalous state in the lower stratosphere can develop or not. This is indeed the key difference between the two

selected ensemble re-forecast simulations with comparably weak SSW events: In SSW 3, the wave reflection surface occurs in the lower stratosphere around 50 hPa and below, whereas SSW 2 develops the reflective surface much higher, namely in the mid to upper stratosphere. The corresponding acceleration of the winds in the lower stratosphere subsequently determines whether anomalies can develop. For simplicity, we illustrated this based on 3 (out of 18) exemplary re-forecasts. We find indications for similar relations between the development of lower stratospheric anomalies with the occurrence / height of

wave reflecting surfaces in many of the 18 ensemble re-forecast simulations, yet substantial differences in the coupled wave-mean-flow evolution make it difficult to deduce an over-arching mechanism.

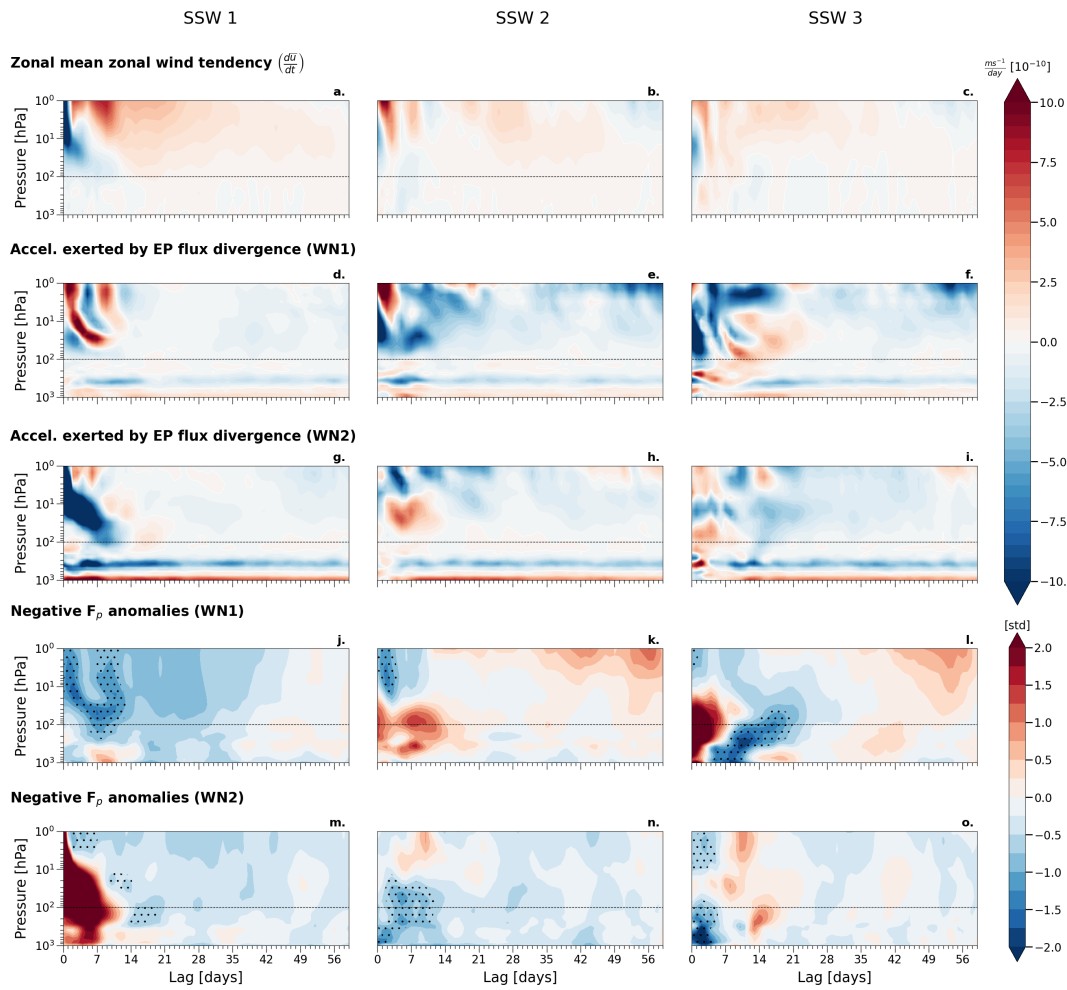

**Figure 14.** Daily means of major terms from the momentum budget of the TEM equations and EP-flux anomalies for the three SSW events shown in Fig. 8. Rows 1-5: zonal mean zonal wind tendency (a-c), acceleration exerted by the EP-flux divergence in WN1 (d-f) and WN2 (g-i), respectively, shown as $ms^{-1}$ per day, and standardised negative EP-flux anomalies in WN1 (j-l) and WN2 (m-o), respectively. Hatchings (j-o) depict regions of downward propagating waves in the ensemble mean, ie. where the ensemble mean EP-flux is positive. All metrics shown here are calculated over the region 45-75°N.





## 8   Discussion

The results presented in the previous section indicate that the height of reflective surfaces could play a role in the coupling
mechanism by determining whether mean flow anomalies associated with an SSW can progress downward into the lower
stratosphere. We further examined both WN1 and WN2 type of events and found no clear distinction in their behaviour in
this context. Overall, and in contrast to some recent studies (e.g. Bett et al., 2023), we do not observe a consistent correlation
between the type of initial event—whether a split or displacement— and the resulting surface impact (not shown), although it
should be noted that the literature on this point is itself inconclusive (Maycock and Hitchcock, 2015, see e.g.,).

Interestingly, Rupp et al. (2022) found for the early 2020 event that the increase in polar vortex strength following the
reflection event influenced the tropospheric flow and thus increased the likelihood of extreme surface values. In other words,
the role of the wave reflection event was to modify the stratospheric mean flow in terms of the polar vortex recovery (and
subsequent strengthening) after the initial sudden stratospheric deceleration event with subsequent downward coupling, rather
than a direct influence of the wave reflection itself on the troposphere. This bears similar tones to our findings, namely that the
wave reflection surfaces here play a determining role in the evolution of the (lower) stratospheric mean flow evolution, which
in turn leads to a surface response. This is a different role of wave reflection in stratosphere-troposphere coupling compared to
studies by e.g. Dunn-Sigouin and Shaw (2020), who discuss a direct downward impact of the wave reflection events themselves.

An inherent aspect of the ensemble re-forecast setup used in this study is that the tropospheric initial conditions are only
slightly perturbed across the ensemble members, leading to initial condition memory within the troposphere over the first 1 to
2 weeks. Consequently, disentangling the relative contributions of initial tropospheric variability and associated wave pulses
to the lower stratospheric evolution outcome remains challenging. However, the absence of a strong correlation between the
initial tropospheric conditions and the later surface response (Fig. 9) suggests that large-scale initial tropospheric circulation
patterns may play a less critical role in determining the eventual surface response following the SSW. It should further be noted
that tropospheric variability emerging after week two can effectively be ruled out as a significant factor in the subsequent lower
stratospheric evolution. One possible approach to disentangle the role of tropospheric versus stratospheric initial condition
memory in determining the development of the lower stratospheric response is to run forecasts with different scrambled initial
states, an example of which can be seen in the work of Davis et al. (2022). They found that forecasts initialized with scrambled
stratospheric initial conditions explained most of the observed surface temperature variability in the month after the SSW.
Further analyses revealed that disturbed stratospheric states may be an important feedback on persistent tropospheric surface
behaviour, rather than its proximate cause.

Examining the longitudinally resolved tropospheric circulation, we find a clear Greenland blocking signal in weeks 3–7
after SSW onset in the re-forecast composite mean (Fig. 12), consistent with earlier case-study work (Kautz et al., 2022,
e.g.). Ensemble members that develop an anomalously strong lower stratosphere two weeks after the initial event show an
amplified surface response, both in the full-field MSLP pattern (Fig. 11) and in blocking frequency (Fig. 12). A further key
finding of our study is illustrated in Fig. 13b, which reveals a strong correlation ($r = 0.84$) between large-scale polar-cap
surface GPH anomalies and regional blocking frequencies over Greenland. Combining the above results and this robust and





near-linear relationship implies a clear sequence of events: *(i)* an anomalous lower-stratospheric state modulates the zonal-mean tropospheric flow and favours a pronounced zonal-mean surface response, which in turn *(ii)* increases the likelihood and intensity of regional atmospheric blocking. For Central Europe this sequence implies a higher probability of cold-air outbreaks (due to the enhanced blocking occurrence over Greenland and the adjacent North Atlantic ocean), while over Scandinavia the simulations still indicate an increased chance of cold spells (not shown), even though they do not reproduce the more frequently discussed Scandinavian-blocking configuration (e.g. Michel et al., 2023; Athanasiadis et al., 2020). Thus, lower stratospheric anomalies act less as a direct trigger and more as a regulator that makes certain tropospheric states, such as Greenland blocking, statistically more likely across the 18 SSW events analysed. Determining the actual mechanism of how the lower stratosphere anomaly couples down to the troposphere will require future studies (see e.g. Baldwin et al., 2021, for an overview of studies on this topic).

Note that the ensemble-mean nature of the re-forecast dataset makes this chain particularly evident; by contrast, the SSW events from the EGE control runs show essentially no correlation between blocking and large-scale surface signals (Fig. 13a), highlighting the added value and statistical robustness of our re-forecast approach.

## 9   Summary and Conclusion

In this study, a dedicated simulation set-up is used to shed light on the question whether, and why, some SSW events are more likely than others to induce a tropospheric response. Firstly, an ensemble of winter simulations provide a set of SSW events under controlled boundary conditions, excluding external factors to cause differences between events. Secondly, ensemble re-forecasts centred around selected SSW events set the stage to ascertain whether downward coupling of SSW events is truly of random nature.

Constructing this simulation setup has two key benefits: i) Using the free-running large-ensemble simulation to generate a large set of realistic wintertime evolutions with SSWs in a controlled climate, while retaining independent stratosphere-troposphere evolution, rules out the possibility that different external forcings (e.g. ENSO, QBO) lead to differences between SSW events in our EGE. In particular, this means that all differences have to result from differences in internal dynamical evolution. ii) The ensemble re-forecasts pave the way for an improved statistical characterisation of the possible tropospheric evolutions following stratospheric warming events, as the quantification of the distribution of the response is facilitated through the ensemble setup. Since the simulations are centred around the start date of the SSW event and hence all 40 members capture the initial SSW event identically, we effectively 'average out' the tropospheric internal variability in the post-event surface response, a key advantage of these ensemble re-forecasts.

In Section 3, we validated the suitability of this EGE for simulating realistic wintertime evolutions by comparing its performance against ERA5 reanalysis data from 1979–2019. Our evaluation showed that the EGE realistically captures enhanced lower-stratospheric and surface anomalies following major SSW events. Notably, the modelled stratosphere-troposphere coupling closely resembles the characteristics of previously observed events reported in observation-based studies (Baldwin and Dunkerton, 2001, e.g.).



To further investigate the variability and predictability of surface responses following SSWs, we conducted ensemble re-
forecast simulations for 18 carefully selected events from the original EGE dataset. These re-forecasts were specifically chosen
to represent a realistic range of variability in initial event strengths, lower-stratospheric anomalies, surface responses, and
occurrence times within the winter season. Our key findings are outlined in the following paragraphs.

Our analysis of the re-forecast simulations revealed substantial event-to-event variability in the likelihood of downward
coupling, i.e., some SSWs are more likely to lead to a tropospheric response than others. Crucially, we identified a strong cor-
relation between the strength of lower-stratospheric anomalies shortly after the SSW onset and the magnitude of the ensemble
average surface signals starting approximately three weeks later. Lower-stratospheric anomalies shortly after the SSW were
found to be superior predictors of subsequent surface anomalies compared to the initial intensity of the warming itself. This
emphasizes the critical role of the lower stratosphere as a mediator controlling the downward coupling process.

Examining the TEM momentum budget highlighted the potential importance of wave reflection events occurring post-
SSW for the mean flow evolution. In particular, the height of the reflective surface can influence whether persistent lower-
stratospheric anomalies are established: if the reflection occurs within the lower stratosphere, the associated acceleration of
mean winds lead to a quick recovery from the SSW-related negative wind anomaly in the lower stratosphere. If the reflection
occurs at higher levels, the lower stratospheric anomalies are less affected and might persist, creating conditions that favour
coupling to the troposphere. By conducting our experiments in with ensembles (averaging out the noise), we isolated this
wave-reflection mechanism and its potential effects on stratosphere-troposphere coupling.

Importantly, we found that individual SSW events exhibit considerable case-dependent differences in their likelihood of
developing the canonical tropospheric response not only in terms of zonal mean circulation, but also on a regional level.
Specifically, stronger polar cap surface anomalies after the warming event translate almost linearly to increased blocking
frequencies over key regions such as Hudson Bay, Greenland, and the North Atlantic Ocean. These differences in SSW response
can be reliably predicted as early as the day of the SSW event itself, with the magnitude of post-event lower-stratospheric
anomalies being an effective predictor.

**Data availability**

This study used the ERA5 re-analysis dataset, details of which can be found on the ECMWF website https://www.ecmwf.
int/en/forecasts/dataset/ecmwf-reanalysis-v5, and be accessed via the Climate Data Store https://cds.climate.copernicus.eu/.
Details pertaining to the ERA5 SSW dataset can be accessed via the Sudden Stratospheric Warming Compendium (https:
//csl.noaa.gov/groups/csl8/sswcompendium/). Datasets including the $\overline{U}_{60N}^{10hPa}$ time series for for numerical ensembles and re-
analysis data will be provided in a public repository upon publication.



**Author contributions**

SL performed the analyses and prepared the manuscript with contributions from all co-authors. PR performed the simulations.
The concept was developed together with all co-authors.

**Competing interests**

Some authors are members of the editorial board of journal WCD.

**Acknowledgements**

The authors thank the Transregional Collaborative Research Center SFB/TRR 165 "Waves to Weather" funded by the German
Research Foundation (DFG) for support. We thank Sebastian Borchert from the Deutscher Wetterdienst (DWD) for providing
helpful insights about the ICON setup.

**Financial support**

This work was supported by the Transregional Collaborative Research Center SFB/TRR 165 "Waves to Weather" funded by
the German Research Foundation (DFG) and the German Aerospace Centre (DLR) in the framework of the Open Access
Publishing Program. JGP thanks the AXA Research Fund for support.



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
