# Peer review of "Case-to-Case Variability in the Tropospheric Response to Sudden Stratospheric Warmings Revealed by Ensemble Re-Forecasts"

_EGUsphere, 2025_

## Referee Comment (RC1)

**Review of 'Case-to-Case Variability in the Tropospheric Response to Sudden Stratospheric Warmings Revealed by Ensemble Re-Forecasts' by Loeffel et al.**

**General comments:**

This study focuses on the variability in the surface response following sudden stratospheric warming (SSW), using the ensemble re-forecasts. The authors identify that the lower stratospheric anomalies can be an effective predictor for the surface response, and attribute the mechanism to the height of the reflective surface. The experiments are nicely designed and the results have the potential to deepen understanding of under which condition an SSW may have strong influence on the surface. However, while the authors highlight the aim of 'quantify' the variable surface response, the analyses are mostly based on ensemble means, so the advantage of using ensemble forecasts is largely masked. Moreover, several strong conclusions are not sufficiently supported by quantitative evidence. In addition, the initial condition setup in the model experiments poses non-negligible limitations and requires further discussion. Hence, I would recommend a major revision, to make the most use of the ensemble forecasts and to provide stronger evidence for both the role of lower-stratospheric anomalies as a predictor and the proposed mechanism. Please see my detailed comments below.

**Major Comments:**

1. Deterministic vs probabilistic analyses. The analyses are mostly based on the ensemble mean. While I understand the authors wanted to rule out the influence of the tropospheric initial condition, the advantage of using ensemble forecasts has been masked. Especially given that the authors highlight the aim of 'quantifying' the variability in the abstract and introduction, this mismatch reduces the significance of this work. More importantly, the key conclusion regarding the role of the lower stratospheric anomalies is based on 18 samples (the ensemble means), which limits the statistical robustness of the results. The proposed mechanism involving the height of reflective surface is based on three samples, which is insufficient for a general conclusion. I suggest the authors make the full use of the large sample size, to better reflect both the uncertainty and the robustness of the results. For example, aside from the scatter plots (e.g., Fig. 10), I recommend the authors stratify the ensemble members for all the selected SSWs based on the lower stratospheric response, and show the temporal evolution of the surface response to check if there is a systematic shift in the distribution of surface response. In this way, it can not only provide stronger evidence to support the role of lower stratospheric condition, but also present a quantitative result on how likely the response would differ (i.e., the probability question arise in the manuscript). Please refer to my specific comments.

2. The interpretation on the role of external variability. The authors highlight that the role of external variability, including ENSO and QBO, has been excluded given that the initial atmospheric condition is the same and the SST has been set to a climatological state. However, the initial conditions from October 2020 possess a specific QBO state, which is then inherent in the model setting for all members. As previous studies have highlighted the role of the QBO influencing the surface response to weak polar vortex events (e.g., Ma et al. 2024), I believe a discussion is needed to better reflect this limitation and contextualize the results here.

Reference:

Ma, J., Chen, W., Yang, R. et al. Downward propagation of the weak stratospheric polar vortex events: the role of the surface arctic oscillation and the quasi-biennial oscillation. Clim Dyn, 62, 4117–4131 (2024). https://doi.org/10.1007/s00382-024-07121-5

3. Manuscript formatting and readability. The manuscript requires a thorough proofreading and correction for formatting and citation errors, which currently detract from the paper's professionalism and readability. For instance, there are several inconsistencies in citation style and grammar, e.g., L363, (Maycock and Hitchcock, 2015, see e.g.,) should be (see e.g., Maycock and Hitchcock, 2015).

**Specific Comments:**

1. L20: This statement on the evolution timescale is ambiguous. The authors seem to be referring to the difference in 'memory' or 'predictability timescales', not just 'evolution timescales', as tropospheric blocking can also persist for weeks.

2. L45-50: Please include the related reference.

3. Section 2: I suggest the authors move the description of the reanalysis dataset to Section 2.1 (i.e., before introducing the model setup). Otherwise, it is unclear what baseline climatology has been used in the model simulation.

4. L76: As mentioned in my major comment 2, the initial atmospheric condition already contains a specific QBO phase. Thus, some discussion should be included, perhaps following L89 where the authors currently highlight that the influence of QBO has been excluded.

5. L96: Does day $d_0$ indicate the calendar day? If so, please clarify this.

6. L120: 'Butler et al. (2015)' should be '(Butler et al., 2015)'

7. L127: Are the results sensitive to the choice of time periods and the threshold? By 'Sensitivity to the choice of time periods in further discussed in Section 5 (see Fig. 9)', I was expecting the authors to show a sensitivity test on these parameters there, but Fig. 9 actually

shows the correlation between the zonal mean circulation with the weeks 3-7 response. A more objective way to define the threshold and period could be to separate the SSWs based on the surface response, and then use the features of the strong surface response group to define the parameters.

8. L164: 'Butler et al. (2017)' should be '(Butler et al., 2017)'.

9. Figure 1: The shading in this plot cannot be seen clearly, especially for the EGE. I suggest modifying the plot for visualization. In addition, I suggest also including the histogram for the ERA5 for a more straightforward comparison between the EGE and reanalysis.

10. L174: 'decreased AO' is not precise, do you mean the 'weakening of a positive AO'?

11. L175: Suggest using 'SSW' consistently, instead of referring to these events as 'weak vortex events' and 'stratospheric event' (L184).

12. Figure 2: How is the climatology defined in these plots, and what is day 0 for the climatology? In addition, the lower panel presents the raw AO index, where an initial positive AO is seen that should be related to the initial tropospheric condition. This might also explain the very noisy evolution of the AO index. Since normally we would expect a negative AO anomaly following an SSW, it might be more straightforward to show the AO anomaly instead. Also, it might be more informative to use shading to present the spread.

13. Figure 3: Suggest showing the plot for the non-LS group as well.

14. Figure 4: How is the climatology defined? Please clarify this.

15. L225: It's a bit awkward to say 'reduction of the zonal circulation', perhaps 'weakening' is clearer.

16. Figure 6: Why are there very positive zonal winds? It is surprising to see the westerly wind during an SSW, is it sensitive to the duration (e.g., two weeks average has been presented here)?

17. L254-255: Are the results sensitive to the selected period (i.e., weeks 3-7)? According to Fig. 9, the correlation coefficient reduces dramatically after around day 30 in the troposphere. This means that the surface response in weeks 6-7 differs from the weeks before. This challenges the selection of using the weeks 3-7 mean.

18. Figure 10: As only the ensemble mean has been shown, it limits the sample size and the robustness of the conclusion. Suggest showing the individual member to better reflect the uncertainty. In addition, Fig. 10b show the linkage between the raw U10 and the surface response. Given the strong seasonal cycle in the stratosphere, the weaker warming (i.e., more positive raw U10) does not necessarily reflect a weaker departure from the climatology. How about the linkage between the anomalous U10 and surface response?

19. L261-264: This question naturally leads the readers to think about the mechanism at this stage. But in the following section, the regional response is shown instead of the mechanism. This narrative flow is a bit disconnected. Also in the abstract and the summary, the mechanism follows the zonal mean circulation response, which is further followed by the regional response. Will this flow work better?

20. L296-297: This is because of they both reflect the intensity change in the stratospheric polar vortex. This statement is confusing. It seems to be stating a known fact from the literature, not a new finding.

21. Section 7: As mentioned in my major comment 1, the analyses in this section are based on three cases, which cannot really represent the general mechanism. I agree that these case study can provide some hints, but a more systematic/objective analysis is needed to further support the hypothesis in the role of wave reflection and height of reflective surface. One possible approach is to calculate the height of the reflective surface, and stratify the members (for all the 18 SSWs) based on the height of the reflective surface, then check the evolution of lower stratospheric anomaly and surface response. If the distribution of these two groups can be distinguished, then it would provide stronger evidence.

22. Figure 14 caption: 'where the ensemble mean EP-flux is positive'. Should it be the ensemble mean vertical component of EP flux is negative?

23. L363. Revise the citation format.

24. L376-377: Figure 9 only presents the linkage between the zonal mean circulation and the surface response, so it cannot really support the strong statement that the initial tropospheric circulation patterns play a less critical role. In fact, in previous study (Ma et al. 2024), the preceding tropospheric circulation is shown be an important factor in influencing the surface response.

Reference:

Ma, J., Chen, W., Yang, R. et al. Downward propagation of the weak stratospheric polar vortex events: the role of the surface arctic oscillation and the quasi-biennial oscillation. Clim Dyn, 62, 4117–4131 (2024). https://doi.org/10.1007/s00382-024-07121-5

25. L391-393: Please add the related reference.

26. L395: A related plot would be helpful. And why do the simulations still indicate an increased chance of cold spells over Scandinavia even though there is not more frequent Scandinavian blocking?

27. L402: What does Fig. 13a refer to? There is only one panel in Fig. 13.

28. L431-432: The correlation coefficient does not really have dramatic difference (r=0.85 vs. r=-0.71). Given the very limited sample size (N=18), it is difficult to draw this strong conclusion.

---

## Author Comment (AC1)

**EGUSPHERE-2025-4164**

Peer Review – Loeffel et al.

'Case-to-Case Variability in the Tropospheric Response to Sudden Stratospheric Warmings Revealed by Ensemble Re-Forecasts'

**Author Responses: First Round**

**Reviewer's Summary**

This study focuses on the variability in the surface response following sudden stratospheric warming (SSW), using the ensemble re-forecasts. The authors identify that the lower stratospheric anomalies can be an effective predictor for the surface response, and attribute the mechanism to the height of the reflective surface. The experiments are nicely designed and the results have the potential to deepen understanding of under which condition an SSW may have strong influence on the surface. However, while the authors highlight the aim of 'quantify' the variable surface response, the analyses are mostly based on ensemble means, so the advantage of using ensemble forecasts is largely masked. Moreover, several strong conclusions are not sufficiently supported by quantitative evidence. In addition, the initial condition setup in the model experiments poses non-negligible limitations and requires further discussion. Hence, I would recommend a major revision, to make the most use of the ensemble forecasts and to provide stronger evidence for both the role of lower-stratospheric anomalies as a predictor and the proposed mechanism. Please see my detailed comments below.

We thank the reviewer for their careful reading of the manuscript and for the constructive and detailed feedback, which we feel has helped improve the clarity, robustness, and presentation of this work. We have addressed all comments on a point-by-point basis below and have revised the manuscript accordingly. Unless stated otherwise, all references to line numbers refer to the revised manuscript without track changes.

In response to the reviewer's Major Comments, we have clarified the experimental design and interpretation of the ensemble framework, strengthened the presentation of uncertainty and robustness, and improved the overall readability and formatting of the manuscript. In particular, we clarify that the study is based on an event-centric ensemble strategy designed to quantify case-to-case variability in stratosphere–troposphere coupling under controlled boundary conditions, rather than on operational forecast skill.

**Major Comments:**

1. Deterministic vs probabilistic analyses. The analyses are mostly based on the ensemble mean. While I understand the authors wanted to rule out the influence of the tropospheric initial condition, the advantage of using ensemble forecasts has been masked. Especially given that the authors highlight the aim of 'quantifying' the variability in the abstract and introduction, this mismatch reduces the significance of this work. More importantly, the key conclusion regarding the role of the lower stratospheric anomalies is based on 18 samples (the ensemble means), which limits the statistical robustness of the results. The proposed mechanism involving the height of reflective surface is based on three samples, which is insufficient for a general conclusion. I suggest the authors make the full use of the large sample size, to better reflect both the uncertainty and the robustness of the results. For example, aside from the scatter plots (e.g., Fig. 10), I recommend the authors stratify the ensemble members for all the selected SSWs based on the lower stratospheric response, and show the temporal evolution of the surface response to check if there is a systematic shift in the distribution of surface response. In this way, it can not only provide stronger evidence to support the role of lower stratospheric condition, but also present a quantitative result on how likely the response would differ (i.e., the probability question arise in the manuscript). Please refer to my specific comments.

The main aim of this study is to test whether different SSWs exhibit distinct tropospheric response strengths. Accordingly, the statements in the abstract and introduction referring to "quantifying variability" concern case-to-case variability across different events, rather than variability across individual ensemble members. The experimental design is therefore intended to provide a robust estimate of each event's tropospheric response strength.

The 40-member spin-off ensembles created for 18 selected SSWs capture the same stratospheric event almost identically during the deterministic phase, while sampling a range of physically consistent post-SSW evolutions thereafter. Within this framework, the ensemble mean represents the response strength to a given SSW event, whereas the ensemble spread reflects internal tropospheric variability. The use of ensemble means therefore does not mask ensemble information, but is a deliberate choice to isolate the stratospheric signal from tropospheric noise: ensemble members are used to filter out tropospheric noise, not to inflate sample size. The effective sample size for these event-to-event relationships is therefore 18 independent SSW events. While 18 may be on the lower end for statistical inference, stratification by lower stratospheric state reveals consistently large and clearly separated responses (see plot below and Fig. 11 in the manuscript), supporting the adequacy of this sample size for the questions addressed here.

We have revised the manuscript in the following ways to more clearly state the intention of our study:

1) Revised the title from "*Case-to-Case Variability in the Tropospheric Response to Sudden Stratospheric Warmings Revealed by Ensemble Re-Forecasts*" to "*Quantifying the tropospheric response to individual sudden stratospheric warmings revealed by an ensemble simulation strategy*".
2) Revised the abstract.
3) Revised the introduction to state more clearly that the tropospheric response can only be inferred from the average of an ensemble, rather than from an individual realisation of the evolution following an SSW event, as any single realisation is strongly influenced by internal tropospheric variability.
4) Revised Fig. 7 so that the boxplots represent the distribution of time-averaged surface anomalies over weeks 3–7, rather than daily values. Using the time-averaged response of each ensemble separates the spread across ensemble members from temporal variations and ensures consistency with the surface-response definition used throughout the manuscript. Importantly, if only a single realisation of an SSW event were available (as in the real atmosphere), the surface response could correspond to any one of the individual ensemble members shown in the boxplot. The ensemble mean therefore provides the most robust estimate of the event's response and is emphasised accordingly in the analysis, for example in lines 276-278:
   > "*At the same time, the broad distributions of the ensemble response confirm that the tropospheric response to an individual SSW cannot be inferred from a single observed realisation: even for SSWs with a strong ensemble-mean response, some individual members exhibit negative time-averaged tropospheric anomalies.*"

Regarding the identification of the role of lower stratospheric anomalies in the tropospheric response, stratifying individual ensemble members—rather than the ensemble mean of the 18 individual SSW events—neither addresses the central question of our study nor provides additional information. The lower stratospheric response during week 2 exhibits little spread across the ensemble members of a given spin-off ensemble, as this period remains within the deterministic phase of the stratospheric evolution (see revised Fig. 10). In contrast, the tropospheric response shows substantial spread (Fig. 7) that is attributable to internal tropospheric variability rather than differences in stratospheric conditions, which are identical across ensemble members by construction. This separation of deterministic stratospheric evolution and stochastic tropospheric variability is precisely why our analysis focuses on ensemble means (see discussion above).

To clarify this point, we have added a measure of ensemble spread as error bars to the scatter plot illustrating the relationship between lower stratospheric anomalies and tropospheric response strength across individual SSW events (Fig. 10). The resulting near-linear correlation demonstrates an even stronger connection than grouping events into binary "LS" and "no-LS" classes. For completeness, we nevertheless include the results of

such grouping below and note that the manuscript does present the corresponding zonally resolved responses for these groups (see Fig. 11).

[Figure]

***Figure R1***. *Composite evolution of the surface response for 18 SSW spin-off events, stratified by the presence (blue) or absence (orange) of a lower stratospheric anomaly in week 2 (LS and noLS, respectively). Shading indicates the spread (standard error of the mean) across the ensemble means of the spin-offs in each group; thickened line segments denote statistical significance at the 95% level. Grey background shading highlights weeks 3–7 following SSW onset. This event-stratified analysis reveals a systematic separation in the subsequent surface geopotential height anomalies over weeks 3–7, providing quantitative support for the modulating role of the lower stratosphere while preserving the event-centric framework of the study.*

With regards to the proposed mechanism involving the height of the reflective surface, we fully agree that three cases are insufficient to establish a general mechanism. As clarified in Section 7, this analysis is intentionally framed as illustrative and hypothesis-generating, rather than as a definitive mechanistic result. While we have examined diagnostics related to wave reflection across all events, substantial case-to-case variability makes it difficult to define a single objective metric (e.g. a fixed reflection height) that would allow a robust stratification of all ensemble members in the manner suggested. We therefore present these results as a hypothesis that serves to motivate future, more targeted studies, and have revised the relevant parts in the manuscript to make this clearer (see below).

2. The interpretation on the role of external variability. The authors highlight that the role of external variability, including ENSO and QBO, has been excluded given that the initial atmospheric condition is the same and the SST has been set to a climatological state. However, the initial conditions from October 2020 possess a specific QBO state, which is then inherent in the model setting for all members. As previous studies have highlighted the role of the QBO influencing the surface response to weak polar vortex events (e.g., Ma et al. 2024), I believe a discussion is needed to better reflect this limitation and contextualize the results here.

Reference:
Ma, J., Chen, W., Yang, R. et al. Downward propagation of the weak stratospheric polar vortex events: the role of the surface arctic oscillation and the quasi-biennial oscillation. Clim Dyn, 62, 4117–4131 (2024). https://doi.org/10.1007/s00382-024-07121-5

Yes: The initial conditions from October 2020 possess a specific QBO state (slightly negative) which is therefore inherent in the model setup for all ensemble members. Our intention was not to imply that the QBO influence is absent from the simulations, but rather that QBO variability across individual SSWs is excluded by

construction. Because all 120 EGE members— and, by extension, all 18 spin-off ensembles— are initialised with the same QBO phase (as well as identical ENSO-related boundary conditions), differences in the subsequent stratospheric or surface evolution cannot arise from differences in QBO- or ENSO-related teleconnections.

To make this distinction explicit and better contextualise our results, we have revised the manuscript accordingly. In particular, we now clarify in the Methods section that all ensemble members are initialised with identical large-scale boundary conditions, including the same QBO phase, such that QBO variability is not sampled across the ensemble. We have also revised the Summary and Conclusions to emphasise that this controlled experimental setup rules out differences in external forcings as a source of event-to-event variability, thereby highlighting the role of internal dynamical evolution under a fixed background state. To support this, we have added the following plots to the Supplement (see Figs. R2 and R3 below).

Lines 114 – 122 (Section 2.1)
>*"Furthermore, the removal of inter-annual variability between ensemble members in the boundary conditions provides a more robust basis for analysing case-to-case differences between individual SSWs and their associated tropospheric responses than would be possible with a (limited) reanalysis dataset. Identical large-scale initial conditions ensure that each winter in the ensemble experiences the same QBO phase (all members drift consistently from a slight westerly into an easterly phase, as measured at 30 hPa - see Fig. S2 in the Supplement). Furthermore, the model does not internally generate an MJO, thus known sources for teleconnections from the tropics are missing or are identical between ensemble members. Consequently, any differences in the surface response between SSW events in our EGE must arise from differences in the internal dynamical evolution of the system, rather than from differences in external or slowly varying forcings outside the mid-latitudes (e.g. ENSO or the QBO).*

Lines 465 – 469 (Summary and Conclusion)
>*"Using the free-running large-ensemble simulation to generate a large set of realistic wintertime evolutions with SSWs in a controlled climate, while retaining independent stratosphere-troposphere evolution, rules out the possibility that differences in external confounding forcings (e.g. ENSO, QBO) lead to differences between SSW events in our EGE. In particular, this means that all differences have to result from differences in internal dynamical evolution."*

[Figure]

**Figure R2**. *EGE composite of zonal-mean zonal wind averaged over 10°S–10°N. Left: Vertical profile at day 0. Right: Time evolution at 30 hPa. The blue line denotes the ensemble mean; shading indicates the ensemble range (minimum–maximum).*

[Figure]

**Figure R3.** *Spin-off composite of zonal-mean zonal wind averaged over 10°S–10°N. Left: Vertical profile at day 0. Right: Time evolution at 30 hPa. The blue line denotes the ensemble mean; shading indicates the ensemble range (minimum–maximum).*

3. Manuscript formatting and readability. The manuscript requires a thorough proofreading and correction for formatting and citation errors, which currently detract from the paper's professionalism and readability. For instance, there are several inconsistencies in citation style and grammar, e.g., L363, (Maycock and Hitchcock, 2015, see e.g.,) should be (see e.g., Maycock and Hitchcock, 2015).

*Thank you for catching this – We apologise for the oversights in the initial version, and have carefully proofread the entire manuscript and corrected citation style, grammar, and formatting issues throughout. This includes the specific example noted by the reviewer, as well as additional instances identified during the revision process.*

**Specific Comments:**

1. L20: This statement on the evolution timescale is ambiguous. The authors seem to be referring to the difference in 'memory' or 'predictability timescales', not just 'evolution timescales', as tropospheric blocking can also persist for weeks.

*We agree that tropospheric flow anomalies can persist for weeks, but these represent exceptions to the typical tropospheric evolution. To clarify that the statement refers to the dominant behaviour and associated predictability timescales, we have revised the sentence to explicitly state that tropospheric dynamics typically evolve on timescales of several days, while stratospheric dynamics evolve on timescales of weeks (line 25):*

> *"Stratospheric and tropospheric dynamics mostly evolve on substantially different timescales, the former typically within weeks, and the latter typically within several days."*

2. L45-50: Please include the related reference.

Owing to the restructuring of the Introduction, the text referred to in lines 45–50 is no longer present in the manuscript. The underlying point has instead been incorporated into a new paragraph explicitly discussing why a quantitative "response" cannot be inferred from a single realisation or event. Appropriate references have been added there (lines 31 – 40).

3. Section 2: I suggest the authors move the description of the reanalysis dataset to Section 2.1 (i.e., before introducing the model setup). Otherwise, it is unclear what baseline climatology has been used in the model simulation.

In our framework, the baseline climatology used to interpret the model results is given by the ensemble mean of the 120-member Event-Generating Ensemble (EGE). The re-analysis dataset is not used to define the model climatology, but rather to provide an independent reference against which the statistical properties of the EGE can be evaluated, demonstrating that the EGE represents a realistic range of wintertime atmospheric states.

For this reason, we intentionally introduce the model setup and the EGE before describing the reanalysis data, as this reflects the logical structure of the methodology and avoids implying that the simulations are constrained or initialised by re-analysis data. We therefore prefer to retain the current ordering of Section 2.

To improve clarity and avoid confusion, we have revised the text to more explicitly state that the model climatology is derived solely from the EGE ensemble mean (e.g. line 95), and that the reanalysis data are used exclusively for evaluation and comparison purposes (lines 147 – 151).

4. L76: As mentioned in my major comment 2, the initial atmospheric condition already contains a specific QBO phase. Thus, some discussion should be included, perhaps following L89 where the authors currently highlight that the influence of QBO has been excluded.

Please see our response to Major Comment 2 above.

5. L96: Does day d0 indicate the calendar day? If so, please clarify this.

Yes, day $d_0$ corresponds to a specific calendar day within the model simulation. More precisely, $d_0$ is defined as the day on which the zonal mean zonal wind at 10 hPa and 60° N first reverses sign and becomes negative, which we define as the onset of the SSW event.

This day therefore corresponds to a specific date (day and month) within the Event-Generating Ensemble (EGE) simulation from which the spin-off ensemble simulation is initialised. However, since the EGE represents idealised wintertime simulations under climatological boundary conditions, this date does not correspond to a specific observed year, but rather to a model-internal calendar date associated with the identified SSW event.

We have clarified this point in the revised manuscript at line 126:
> "Here, $d_0$ denotes the onset date of the SSW, defined as the first day on which the zonal-mean zonal wind at 10 hPa and 60 °N becomes negative."

6. L120: 'Butler et al. (2015)' should be '(Butler et al., 2015)

Thank you for catching this – the citation format has been corrected.

7. L127: Are the results sensitive to the choice of time periods and the threshold? By 'Sensitivity to the choice of time periods in further discussed in Section 5 (see Fig. 9)', I was expecting the authors to show a sensitivity test on these parameters there, but Fig. 9 actually shows the correlation between the zonal mean circulation with the weeks 3-7 response. A more objective way to define the threshold and period could be to separate the SSWs based on the surface response, and then use the features of the strong surface response group to define the parameters.

The lower stratospheric response is defined using the event-wise ensemble-mean polar-cap geopotential height anomaly at 100 hPa averaged over days 7–14 after SSW onset, setting a threshold of +1.5 standard deviations. This period corresponds to the early post-SSW adjustment phase, during which differences in the subsequent lower-stratospheric evolution first emerge across events. Using a time-averaged signal over this window reduces sensitivity to short-lived fluctuations and provides a stable discriminator between events with and without a pronounced lower-stratospheric response. The tropospheric response is defined as the ensemble-mean polar-cap geopotential height anomaly at 1000 hPa averaged over weeks 3–7 after SSW onset.

The statement that "sensitivity to the choice of time periods [is discussed in Fig. 9]" was perhaps misleading. Figure 9 is intended as a diagnostic of the temporal evolution of correlations, rather than as a formal sensitivity test, and motivates the choice of the averaging window. We have corrected this wording to clarify that Fig. 9 motivates the choice of the averaging window but does not represent a parameter-tuning exercise in lines 167 – 172:

> *"These time ranges are motivated by the timescales typically associated with the downward propagation of stratospheric mean-flow anomalies, spanning days to approximately two months (see e.g., Ding et al., 2023; Scaife et al., 2022; Sigmond et al., 2013), and are further supported by the composite results of this study (e.g. Figs. 3 and 8). We tested the sensitivity to the exact definition of these time ranges, and results are generally robust with respect to variations of a few weeks (not shown). The weeks 3–7 period is therefore chosen to capture the delayed sub-seasonal surface response following SSW onset, while allowing for event-to-event differences in response timing."*

8. L164: 'Butler et al. (2017)' should be '(Butler et al., 2017)'.

Thank you for catching this – the citation format has been corrected.

9. Figure 1: The shading in this plot cannot be seen clearly, especially for the EGE. I suggest modifying the plot for visualization. In addition, I suggest also including the histogram for the ERA5 for a more straightforward comparison between the EGE and reanalysis.

We agree and have revised Figure 1 to improve the visibility of the shading, particularly for the EGE distribution. In addition, we have included the corresponding histogram for ERA5 to allow for a more direct visual comparison between the EGE and reanalysis.

10. L174: 'decreased AO' is not precise, do you mean the 'weakening of a positive AO'?

Yes – The sentence refers to the difference between the EGE climatology and the SSW composite shown in Fig. 2b, where the AO according to our proxy index remains positive but is weakened following SSW onset. In response to other comments (see below), we have now revised the figure to show standardised anomalies (more in line with the standard definition of the AO), and have changed the text accordingly (line 215):

> *"This tropospheric signal of negative ÃO index anomalies remains statistically significant for up to five to six weeks following the SSW onset"*

11. L175: Suggest using 'SSW' consistently, instead of referring to these events as 'weak vortex events' and 'stratospheric event' (L184).

We agree that the term "weak vortex events" can be ambiguous and may encompass situations beyond formally defined SSWs. We have therefore revised the manuscript to avoid this terminology and to consistently refer to these events as SSWs where appropriate.

In some places, we retain the more general term "stratospheric event" for stylistic reasons and to improve sentence flow, while still referring unambiguously to SSWs within the surrounding context. We have checked the text throughout to ensure that the terminology is used consistently and without ambiguity.

12. Figure 2: How is the climatology defined in these plots, and what is day 0 for the climatology? In addition, the lower panel presents the raw AO index, where an initial positive AO is seen that should be related to the initial tropospheric condition. This might also explain the very noisy evolution of the AO index. Since normally we would expect a negative AO anomaly following an SSW, it might be more straightforward to show the AO anomaly instead. Also, it might be more informative to use shading to present the spread.

In Fig. 2, the climatology is defined as the ensemble mean of the 120-member Event-Generating Ensemble (EGE). For the climatological reference, day 0 corresponds to randomly selected winter dates (January–February) within each EGE member, such that the time axis represents lag days relative to a random reference date. This is now stated explicitly in lines 216-218:

> *"For the climatological reference shown in Fig. 2, day 0 corresponds to randomly selected winter dates (January–February) within each EGE member, such that the time axis represents lag days relative to a random reference date."*

To make the post-SSW signal clearer and more consistent with common practice, we have revised Fig. 2 to show the standardised $\tilde{AO}$ anomaly relative to the EGE climatology, rather than the raw index. This more directly highlights the expected negative $\tilde{AO}$ signal following SSW onset. We have also revised Fig. 8 accordingly. In addition, we now represent ensemble spread with shading rather than individual ensemble members, improving figure readability.

13. Figure 3: Suggest showing the plot for the non-LS group as well.

To maintain brevity in the main text, we have chosen not to further expand Figure 3 in the manuscript. For completeness, the corresponding plot for the non-LS group has been added to the supplementary material and is referenced in the main text.

14. Figure 4: How is the climatology defined? Please clarify this.

In Fig. 4, the climatology is defined as the ensemble mean of the 120-member Event-Generating Ensemble (EGE). All anomalies shown in the figure are computed relative to this EGE climatology. This definition has now been stated explicitly in the revised figure caption and accompanying text.

15. L225: It's a bit awkward to say 'reduction of the zonal circulation', perhaps 'weakening' is clearer.

We agree that "weakening" is clearer in this context and have revised the sentence accordingly (line 266).

16. Figure 6: Why are there very positive zonal winds? It is surprising to see the westerly wind during an SSW, is it sensitive to the duration (e.g., two weeks average has been presented here)?

Yes, the positive zonal winds in Fig. 6 arise from the temporal averaging applied in the analysis. The figure shows a two-week mean zonal-mean zonal wind following SSW onset, rather than instantaneous values at day zero.

While SSWs are defined by a reversal of the zonal-mean zonal wind at 10 hPa and 60° N, the post-SSW evolution differs substantially across events, with some exhibiting rapid recovery and strengthening of the vortex, while others recover more slowly. Averaging over a two-week period therefore incorporates post-SSW recovery behaviour (rather than the wind reversal itself), allowing positive (westerly) winds to appear in the ensemble mean.

We have clarified this in the revised manuscript by updating the figure axis label from "Initial U1060" to "U1060" and by explicitly stating the applied averaging period in the figure caption.

17. L254-255: Are the results sensitive to the selected period (i.e., weeks 3-7)? According to Fig. 9, the correlation coefficient reduces dramatically after around day 30 in the troposphere. This means that the

surface response in weeks 6-7 differs from the weeks before. This challenges the selection of using the weeks 3-7 mean.

*The weeks 3–7 window is used to characterise the (later) tropospheric surface response. This choice is guided by both prior literature and our diagnostics: ensemble-mean surface responses typically emerge from weeks 3–5 onward, while individual spin-off events can exhibit persistent responses extending into weeks 6–7 (see examples shown in Fig. 8). The weeks 3–7 mean therefore captures the dominant sub-seasonal response while avoiding undue sensitivity to decorrelation at later times.*

*We found that using slightly shorter averaging windows (e.g. weeks 3–5 or 3–6) does not qualitatively alter the results; however, extending the window to week 7 avoids excluding events with later-developing surface responses, as illustrated by individual spin-off evolutions. We have revised the text to clarify this motivation and to correct the statement that sensitivity to the chosen period is "discussed in Fig. 9," which instead serves as a diagnostic of the temporal evolution of correlations rather than a formal sensitivity test.*

*We have added the following sentence where weeks 3–7 is first introduced, in lines 169 – 172 :*

> *"We tested the sensitivity to the exact definition of these time ranges, and results are generally robust with respect to variations of a few weeks. The weeks 3–7 period is therefore chosen to capture the delayed sub-seasonal surface response following SSW onset, while allowing for event-to-event differences in response timing."*

18. Figure 10: As only the ensemble mean has been shown, it limits the sample size and the robustness of the conclusion. Suggest showing the individual member to better reflect the uncertainty. In addition, Fig. 10b show the linkage between the raw U10 and the surface response. Given the strong seasonal cycle in the stratosphere, the weaker warming (i.e., more positive raw U10) does not necessarily reflect a weaker departure from the climatology. How about the linkage between the anomalous U10 and surface response?

*As detailed in our response to Major Comment 1, the ensemble mean shown in Fig. 10 does not represent a reduction in sample size in our framework. Each data point corresponds to the ensemble-mean response of a 40-member spin-off ensemble associated with a single SSW event, and thus represents the mean tropospheric response to that event. Individual ensemble members are not independent realisations of different events, but instead sample internal tropospheric variability under identical stratospheric conditions; showing them would therefore not add independent information relevant to the event-to-event relationship shown in Fig. 10.*

*To better convey uncertainty, we have revised Fig. 10 to include standard errors of the ensemble mean as error bars. These demonstrate that the differences in surface response across SSW events are robust and not an artefact of internal variability, and that the lower stratospheric signal exhibits little spread across ensemble members During the deterministic phase.*

*Regarding the second point, we agree that the strong seasonal cycle of the stratospheric zonal-mean flow is an important consideration when comparing raw wind values. Because all events are conditioned on SSW onset and occur within the winter season, variations in U10/60 in Fig. 10 primarily reflect differences in SSW strength across events rather than a departure from a fixed seasonal climatology.*

*We note, however, that some weak residual seasonality may still influence post-onset vortex recovery, as radiative tendencies to restore the polar vortex vary over the winter season. Importantly, SSWs are identified based on an absolute wind reversal (U10/60 < 0), which constitutes a more stringent criterion earlier in winter; early-season events are therefore, if anything, disadvantaged in generating large post-SSW zonal-mean zonal wind values (U10/60). We have clarified this interpretation in the revised manuscript to avoid confusion.*

19. L261-264: This question naturally leads the readers to think about the mechanism at this stage. But in the following section, the regional response is shown instead of the mechanism. This narrative flow is a bit disconnected. Also in the abstract and the summary, the mechanism follows the zonal mean circulation response, which is further followed by the regional response. Will this flow work better?

Your comment raises an important point regarding the narrative flow of the manuscript. We agree that the question posed in lines 261–264 naturally motivates a discussion of potential mechanisms, and we appreciate your suggestion regarding alternative structuring.

We have deliberately chosen to present the regional tropospheric response before discussing possible mechanisms. The primary aim of the regional-response section is to complete the presentation of the large-scale and regional circulation signals associated with the stratospheric evolution, while the subsequent mechanism-oriented section is intended as a pre-discussion rather than a comprehensive results section. By placing the mechanism discussion toward the end of the Results, immediately preceding the Discussion section, we aim to clearly signal that these mechanisms are hypotheses and interpretations, rather than definitive conclusions.

Regarding the abstract and summary, we agree that the thematic sequence differs slightly. There, the ordering reflects a conceptual stratosphere–stratosphere–troposphere narrative, which we believe is appropriate for summarising the study at a high level. In the main text, however, we prefer to first establish the robustness of the zonal-mean and regional responses before introducing possible mechanisms.

To reduce potential confusion, we have revised the transition text to better guide the reader through this structure and to clarify the interpretative nature of the mechanism discussion:

Lines 310 – 313:

> *"Given that the strength of the lower stratospheric anomaly is of critical importance, but not uniquely linked to the SSW strength itself, the question arises which dynamical mechanisms lead to (or prevent) the development of the lower stratospheric anomaly. While we do not intend to fully answer this question, we explore possible mechanisms in Section 7. Before, we turn to the question whether the distinct zonal mean tropospheric response also transfers to distinct regional impacts."*

20. L296-297: This is because of they both reflect the intensity change in the stratospheric polar vortex. This statement is confusing. It seems to be stating a known fact from the literature, not a new finding.

We agree that the close relationship between zonal-mean zonal wind and GPH anomalies as measures of polar vortex strength is well established in literature. Our intention was not to present this relationship as a new finding, but to demonstrate that, within our framework, the lower-stratospheric wind and GPH anomalies provide a consistent characterisation of the evolving vortex state. Analysing changes in zonal-mean zonal wind thereafter serves to explain the differences found in the GPH anomalies.

We have revised the text to accordingly to clarify this point and to emphasise that the strong correlation is reported here to motivate the subsequent mechanistic analysis, rather than as a novel result (e.g., revised lines 356 – 359):
> *"Consistent with the established understanding of polar vortex dynamics, we find that the lower-stratospheric zonal-mean zonal wind (at 100 hPa) is strongly correlated ($r = -0.91$) with GPH anomalies at the same level (not shown), indicating that both diagnostics provide a coherent measure of the evolving vortex state in our framework."*

21. Section 7: As mentioned in my major comment 1, the analyses in this section are based on three cases, which cannot really represent the general mechanism. I agree that these case study can provide some hints, but a more systematic/objective analysis is needed to further support the hypothesis in the role of wave reflection and height of reflective surface. One possible approach is to calculate the height of the reflective surface, and stratify the members (for all the 18 SSWs) based on the height of the reflective surface, then check the evolution of lower stratospheric anomaly and surface response. If the distribution of these two groups can be distinguished, then it would provide stronger evidence.

We agree with the reviewer that a fully systematic and objective analysis across all 18 SSW events would be required to establish a general mechanism linking wave reflection, the height of the reflective surface, and the

subsequent development of lower stratospheric and surface anomalies. We also agree that the approach suggested by the reviewer— quantifying the height of the reflective surface and stratifying events accordingly— is a natural and promising avenue for future work.

In the present study, however, the analyses in Section 7 are deliberately framed as illustrative and hypothesis-generating. As stated in the manuscript (see below), we focus on three representative ensemble spin-off simulations (out of 18) to highlight the contrast between cases in which the reflective surface occurs in the lower stratosphere (around 50 hPa and below) and cases in which it is located higher in the mid-to-upper stratosphere. These examples are chosen to demonstrate how differences in the vertical structure of wave reflection can be associated with markedly different lower stratospheric wind evolutions following SSW onset.

We have examined diagnostics related to wave reflection and its vertical structure across a broader subset of the 18 SSW events. While we find qualitative indications that similar relationships may hold more generally, the coupled wave–mean-flow evolution exhibits substantial case-to-case variability and strong temporal dependence. In particular, the height and structure of the reflective surface can evolve rapidly in time and are not always well described by a single characteristic level, making it difficult to define an objective, event-independent metric for stratifying all events in the manner suggested.

For this reason, we intentionally refrain from presenting a systematic classification as a definitive result and instead present the analysis in Section 7 as a conceptual exploration of possible mechanisms, rather than a comprehensive mechanistic proof. We have revised Section 7 accordingly to clarify this scope, e.g.:
Lines 347 – 350:
> *"In the following, we discuss possible mechanisms that might prevent the development of lower stratospheric anomalies on the example of three SSW events. While we have examined diagnostics related to wave reflection across a broader set of SSW events, substantial case-to-case variability and strong temporal dependence make it difficult to define a single objective metric for a systematic classification; we therefore focus here on three representative cases to illustrate possible mechanisms."*

and lines 408 – 413:
> *"While this mechanism is illustrated here using three exemplary spin-off simulations (out of 18), we find indications of similar relationships between lower stratospheric anomaly development and the occurrence and height of wave reflection surfaces in additional cases. However, substantial case-to-case differences in the coupled wave-mean-flow evolution suggest that there may not be a single universal mechanism. We therefore leave it to future studies to further investigate the role and generality of the wave reflection mechanism suggested by these examples."*

22. Figure 14 caption: 'where the ensemble mean EP-flux is positive'. Should it be the ensemble mean vertical component of EP flux is negative?

Yes, this was an error in the caption. We have corrected it to state that the ensemble-mean vertical component of the EP flux is negative.

23. L363. Revise the citation format.

Thank you for catching this – the citation format has been corrected.

24. L376-377: Figure 9 only presents the linkage between the zonal mean circulation and the surface response, so it cannot really support the strong statement that the initial tropospheric circulation patterns play a less critical role. In fact, in previous study (Ma et al. 2024), the preceding tropospheric circulation is shown be an important factor in influencing the surface response.
Reference:
Ma, J., Chen, W., Yang, R. et al. Downward propagation of the weak stratospheric polar vortex events: the role of the surface arctic oscillation and the quasi-biennial oscillation. Clim Dyn, 62, 4117–4131 (2024). https://doi.org/10.1007/s00382-024-07121-5

Fig. 9 shows the Pearson correlation between polar-cap geopotential height anomalies at different pressure levels and the 1000 hPa polar-cap geopotential height anomalies averaged over weeks 3–7. The absence of a strong correlation in the troposphere at early lead times therefore indicates that large-scale initial tropospheric circulation anomalies, as defined in this framework, are only weakly related to the subsequent surface response in the ensemble mean.

We agree that this does not preclude an important role of tropospheric processes more generally, nor does it contradict previous studies based on reanalysis and real events (e.g. Ma et al., 2024), which highlight the influence of specific tropospheric circulation patterns on surface impacts.

An important distinction is that such studies include the full range of coupled variability and external forcings (e.g. SST anomalies, QBO-related influences), whereas our experimental design deliberately holds these boundary conditions fixed. As a result, tropospheric precursor signals associated with these forcings are excluded by construction, which likely contributes to the weak tropospheric correlations seen in Fig. 9.

Rather, our result suggests that, within this controlled, event-centric ensemble framework, large-scale initial tropospheric anomalies alone are not a strong predictor of the later surface response when compared to the evolving lower stratospheric state. To avoid overstatement, we have revised the corresponding sentence to more clearly reflect this limited and conditional interpretation:
Lines 432 – 437:

> *"However, the absence of a strong correlation in Fig. 9 suggests that, within this experimental framework, large-scale initial tropospheric circulation anomalies, as defined here, are only weakly related to the subsequent surface response following an SSW. This is consistent with our experimental design, which excludes externally forced tropospheric anomalies (e.g. SST-related signals), such that potential tropospheric precursor signals are absent by construction and the initial tropospheric state is therefore not a decisive factor for the subsequent stratosphere–troposphere evolution."*

25. L391-393: Please add the related reference.

The intention of this sentence was to refer to the robustness of the relationship within our results, rather than to a general result established in the literature. We have revised the wording to make this explicit and to avoid the impression that an external reference is required.
Lines 448 – 449:

> *"Combining the above results, the robust and near-linear relationship identified here implies a clear sequence of events: [...]"*

26. L395: A related plot would be helpful. And why do the simulations still indicate an increased chance of cold spells over Scandinavia even though there is not more frequent Scandinavian blocking?

The relationship between regional blocking, large-scale circulation anomalies, and surface temperature extremes— particularly over Scandinavia— is indeed complex and cannot be inferred robustly from the diagnostics presented here.

As this aspect is not central to the main focus of the paper and would require additional targeted analyses to support a meaningful interpretation, we have chosen to remove this statement from the revised manuscript rather than introduce an additional plot that could invite overinterpretation. The revised text now focuses on circulation-based results that are directly supported by the analyses presented.

27. L402: What does Fig. 13a refer to? There is only one panel in Fig. 13.

Thank you for catching this; this was a typographical error. The reference has been corrected to Fig. 13, and the panel label has been removed.

28. L431-432: The correlation coefficient does not really have dramatic difference (r=0.85 vs. r=-0.71). Given the very limited sample size (N=18), it is difficult to draw this strong conclusion.

Agreed. Given the limited sample size (N = 18), we have revised the text to avoid overinterpreting differences in correlation strength. The sentence has been modified to reflect that both relationships indicate a strong association, without implying a dramatic distinction between them (revised lines 486 – 488):

> "Lower stratospheric anomalies shortly after the SSW tend to show a stronger relationship with subsequent surface anomalies than the initial intensity of the warming itself (based on correlations across the 18 event-wise ensemble-mean responses)."